# Hypoxia truncates and constitutively activates the key cholesterol synthesis enzyme squalene monooxygenase

Hudson W Coates[1], Isabelle M Capell-Hattam[1], Ellen M Olzomer[1], Ximing Du[1], Rhonda Farrell[2,3], Hongyuan Yang[1], Frances L Byrne[1], Andrew J Brown[1]*

[1]School of Biotechnology and Biomolecular Sciences, UNSW Sydney, Sydney, Australia; [2]Prince of Wales Private Hospital, Randwick, Australia; [3]Chris O'Brien Lifehouse, Camperdown, Australia

**Abstract** Cholesterol synthesis is both energy- and oxygen-intensive, yet relatively little is known of the regulatory effects of hypoxia on pathway enzymes. We previously showed that the rate-limiting and first oxygen-dependent enzyme of the committed cholesterol synthesis pathway, squalene monooxygenase (SM), can undergo partial proteasomal degradation that renders it constitutively active. Here, we show hypoxia is a physiological trigger for this truncation, which occurs through a two-part mechanism: (1) increased targeting of SM to the proteasome via stabilization of the E3 ubiquitin ligase MARCHF6 and (2) accumulation of the SM substrate, squalene, which impedes the complete degradation of SM and liberates its truncated form. This preserves SM activity and downstream pathway flux during hypoxia. These results uncover a feedforward mechanism that allows SM to accommodate fluctuating substrate levels and may contribute to its widely reported oncogenic properties.

*For correspondence:
aj.brown@unsw.edu.au

## Editor's evaluation

Cholesterol biosynthesis is a highly oxygen-intensive process as the synthesis of one molecule of cholesterol consumes 11 molecules of oxygen. This valuable paper provides a new link between oxygen sensing and cholesterol synthesis by showing that under conditions of hypoxia (oxygen deprivation), a key cholesterol synthesis enzyme called squalene monooxygenase (SM) is partially degraded to a truncated form that is constitutively active. The supporting evidence is solid and suggests that unregulated activation of SM under oxygen-deficient conditions could reduce the toxicity of squalene and other sterol intermediates.

## Introduction

Cholesterol is an essential component of mammalian cell membranes, yet its aberrant accumulation is detrimental (*Baigent et al., 2010*). Most cellular cholesterol arises from an energetically expensive biosynthetic pathway requiring eleven oxygen molecules and over one hundred ATP equivalents per molecule of product (*Brown et al., 2021*). Furthermore, many intermediates of this pathway are toxic in excess (*Porter and Herman, 2011*). Coordinated regulation of cholesterol synthesis enzymes is therefore vital to ensure the pathway is active only when required, and sufficient substrates and cofactors are available to maintain flux through the full length of the pathway.

Squalene monooxygenase (SM, also known as squalene epoxidase or SQLE, EC:1.14.14.17) catalyzes the rate-limiting conversion of squalene to monooxidosqualene in the committed cholesterol synthesis pathway (*Gill et al., 2011*; *Chua et al., 2020*). This reaction is the first in the pathway to

**eLife digest** Cells need cholesterol to work properly but too much cholesterol is harmful and can contribute to atherosclerosis (narrowing of blood vessels), cancer and other diseases. Cells therefore carefully control the activity of the enzymes that are involved in making cholesterol, including an enzyme known as squalene monooxygenase.

When the level of cholesterol in a cell rises, a protein called MARCHF6 adds molecules of ubiquitin to squalene monooxygenase. These molecules act as tags that direct the enzyme to be destroyed by a machine inside cells, known as the proteasome, thereby preventing further (unnecessary) production of cholesterol.

Previous studies found that squalene monooxygenase is sometimes only partially broken down to make a shorter (truncated) form of the enzyme that is permanently active, even when the level of cholesterol in the cell is high. However, it was unclear what triggers this partial breakdown.

The process of making cholesterol uses a lot of oxygen, yet many cancer cells thrive in tumours with low levels of oxygen. Here, Coates et al. used biochemical and cell biology approaches to study the effect of low oxygen levels on the activity of squalene monooxygenase in human cells. The experiments revealed that low oxygen levels trigger squalene monooxygenase to be partially degraded to make the truncated form of the enzyme.

Firstly, MARCHF6 accumulates and adds ubiquitin to the enzyme to accelerate its delivery to the proteasome. Secondly, as the proteasome starts to degrade the enzyme, a build-up of squalene molecules impedes further breakdown of the enzyme. This mechanism preserves squalene monooxygenase activity when oxygen levels drop in cells, which may compensate for temporary oxygen shortfalls and allow cells to continue to make cholesterol.

Squalene monooxygenase is overactive in individuals with a wide variety of diseases including fatty liver and prostate cancer. Drugs that block squalene monooxygenase activity have been shown to stop cancer cells from growing, but unfortunately these drugs are also toxic to mammals. These findings suggest that reducing the activity of squalene monooxygenase in more subtle ways, such as stopping it from being partially degraded, may be a more viable treatment strategy for cancer and other diseases associated with high levels of cholesterol.

require molecular oxygen, with the introduced epoxide group ultimately forming the signature C3-hydroxyl group of cholesterol. SM can also act a second time on monooxidosqualene to produce dioxidosqualene, the precursor of the potent regulatory oxysterol 24(S),25-epoxycholesterol (*Bai et al., 1992*). As a flux-controlling enzyme, SM is subject to metabolic regulation at both the transcriptional level via sterol regulatory element-binding proteins (*Horton et al., 2003*) and the post-translational level via ubiquitination and proteasomal degradation (*Gill et al., 2011*). The latter is mediated by the N-terminal regulatory domain of SM (SM-N100), which senses lipid levels in the endoplasmic reticulum (ER) membrane and accelerates or attenuates SM degradation in response to excess cholesterol or squalene, respectively (*Chua et al., 2017*; *Yoshioka et al., 2020*). These reciprocal feedback and feedforward loops fine-tune SM activity according to metabolic supply and demand. SM is typically fully degraded by the proteasome; however, incomplete proteolysis produces a truncated form of SM (trunSM) that lacks a large portion of the lipid-sensing SM-N100 domain but retains the full catalytic domain (*Coates et al., 2021*). This renders trunSM cholesterol-resistant and therefore constitutively active. Although truncation is induced by the SM inhibitor NB-598, human cell lines express similar levels of full-length and truncated SM (*Coates et al., 2021*). This points to the existence of an unknown physiological trigger for truncation.

Clarifying the mechanisms of SM regulation is particularly pertinent given the importance of the enzyme, and cholesterol more generally (*Kuzu et al., 2016*), in oncogenesis. Overexpression of the SM gene *SQLE* is associated with greater invasiveness and lethality in breast (*Brown et al., 2016*), prostate (*Stopsack et al., 2016*; *Pudova et al., 2020*), and pancreatic cancers (*Bai et al., 2021*), amongst others. At the protein level, aberrant SM expression is implicated in colorectal cancer progression (*He et al., 2021*; *Jun et al., 2021*) and the development of both nonalcoholic steatohepatitis and hepatocellular carcinoma (*Liu et al., 2021*; *Liu et al., 2018*). Given its key role in oxygen-dependent cholesterol synthesis, SM may be particularly critical for cancer cell survival during hypoxia, which is common

in the poorly vascularized cores of solid tumors and often associated with poor prognosis (*Rankin and Giaccia, 2016*). In support of this idea, SM inhibition sensitizes breast and colorectal cancer cells to hypoxia-induced cell death (*Haider et al., 2016*). Although hypoxic cells tend to accumulate cholesterol, there are conflicting reports on changes in biosynthetic flux (*Mukodani et al., 1990*; *Parathath et al., 2011*; *Wu et al., 2020*). Furthermore, with the notable exception of the early pathway enzyme 3-hydroxy-3-methylglutaryl-CoA reductase (HMGCR) (*Nguyen et al., 2007*), the effects of hypoxia on individual biosynthetic enzymes are unknown. It is also unclear if these might be perturbed in a tumor context to favor continued cholesterol synthesis and cell proliferation.

Here, we show hypoxic conditions induce SM truncation in a variety of cell lines through a combination of accelerated proteasomal degradation and inhibition of its complete proteolysis. This occurs due to the accumulation of both MARCHF6, the major E3 ubiquitin ligase for SM, and squalene, which impedes SM degradation through a mechanism involving the SM-N100 regulatory domain. Taken together, our findings point towards a role for the constitutively active trunSM in adaptations to hypoxic conditions and suggest it may contribute to the oncogenic impacts of SM activity.

## Results

### Oxygen availability regulates SM truncation

We previously showed that SM is post-translationally regulated by its substrate squalene and pathway end-product cholesterol (*Gill et al., 2011*; *Yoshioka et al., 2020*). The enzyme also undergoes partial proteasomal degradation of its N-terminus to liberate a truncated protein (trunSM) that is cholesterol-resistant and thus constitutively active (*Figure 1A*; *Coates et al., 2021*), although physiological triggers are unknown. As SM is a rate-limiting enzyme of cholesterol synthesis and catalyzes its first oxygen-dependent reaction (*Figure 1—figure supplement 1*), we tested if SM protein levels are affected by oxygen availability. Incubation of HEK293T cells under hypoxic conditions (1% $O_2$) stabilized hypoxia-inducible factor-1α (HIF1α; *Figure 1B*) and upregulated its target genes *VEGF* and *CA9* (*Figure 1—figure supplement 2A*), confirming the induction of a hypoxic response. We also noted a striking increase in SM truncation caused by the disappearance of full-length SM and a four-fold accumulation of trunSM (*Figure 1B*). This led to trunSM becoming the predominant SM variant, as indicated by the elevated trunSM:SM ratio (*Figure 1—figure supplement 2B*). Hypoxia-induced truncation of SM increased over time (*Figure 1C*, *Figure 1—figure supplement 2C*) and according to the magnitude of oxygen deprivation (*Figure 1D*, *Figure 1—figure supplement 2D*), with the net result of increased total enzyme levels (expressed as the sum of full-length SM and trunSM levels; *Figure 1—figure supplement 1C*). Importantly, trunSM accumulation was greater under the severely hypoxic conditions characteristic of solid tumors (0.5–2% $O_2$) than the 'physoxic' conditions experienced by normal human tissues in situ (3–7.5% $O_2$) (*Figure 1D*, *Figure 1—figure supplement 2D*; *McKeown, 2014*). This suggested that increased SM truncation is a feature of pathophysiological hypoxia. We also noted our experiments, which for technical reasons used a variety of cell seeding densities, showed variation in the normoxic trunSM:SM ratio. Indeed, we confirmed SM truncation is increased at higher cell densities and accompanied by slight stabilization of HIF1α (*Figure 1—figure supplement 2E*), consistent with other reports (*Sheta et al., 2001*; *Dayan et al., 2009*). This further supported the phenomenon of hypoxia-induced truncation.

We also surveyed SM levels in a panel of cell lines and found hypoxia-induced accumulation of trunSM was generalizable to all, although full-length SM levels did not decline in MDA-MB-231 breast cancer cells (*Figure 1—figure supplement 2F*). As HIF1α and hypoxia-inducible factor-2α (HIF2α) transcriptionally regulate the cellular response to hypoxia, we next tested if their activity is required for SM truncation. However, knockdown of *HIF1A* and *HIF2A* expression to a level sufficient to reduce target gene activation (*Sena et al., 2014*) had no effect on the magnitude of hypoxia-induced SM truncation in HEK293T cells (*Figure 1—figure supplement 3A, B*). This ruled out the involvement of HIF1α, HIF2α and their target genes in this phenomenon.

### Hypoxia transcriptionally and post-translationally reduces full-length SM levels

As SM is truncated via partial proteasomal degradation (*Coates et al., 2021*), we reasoned that hypoxia promotes this through a two-step mechanism: (1) targeting of full-length SM to the proteasome, and

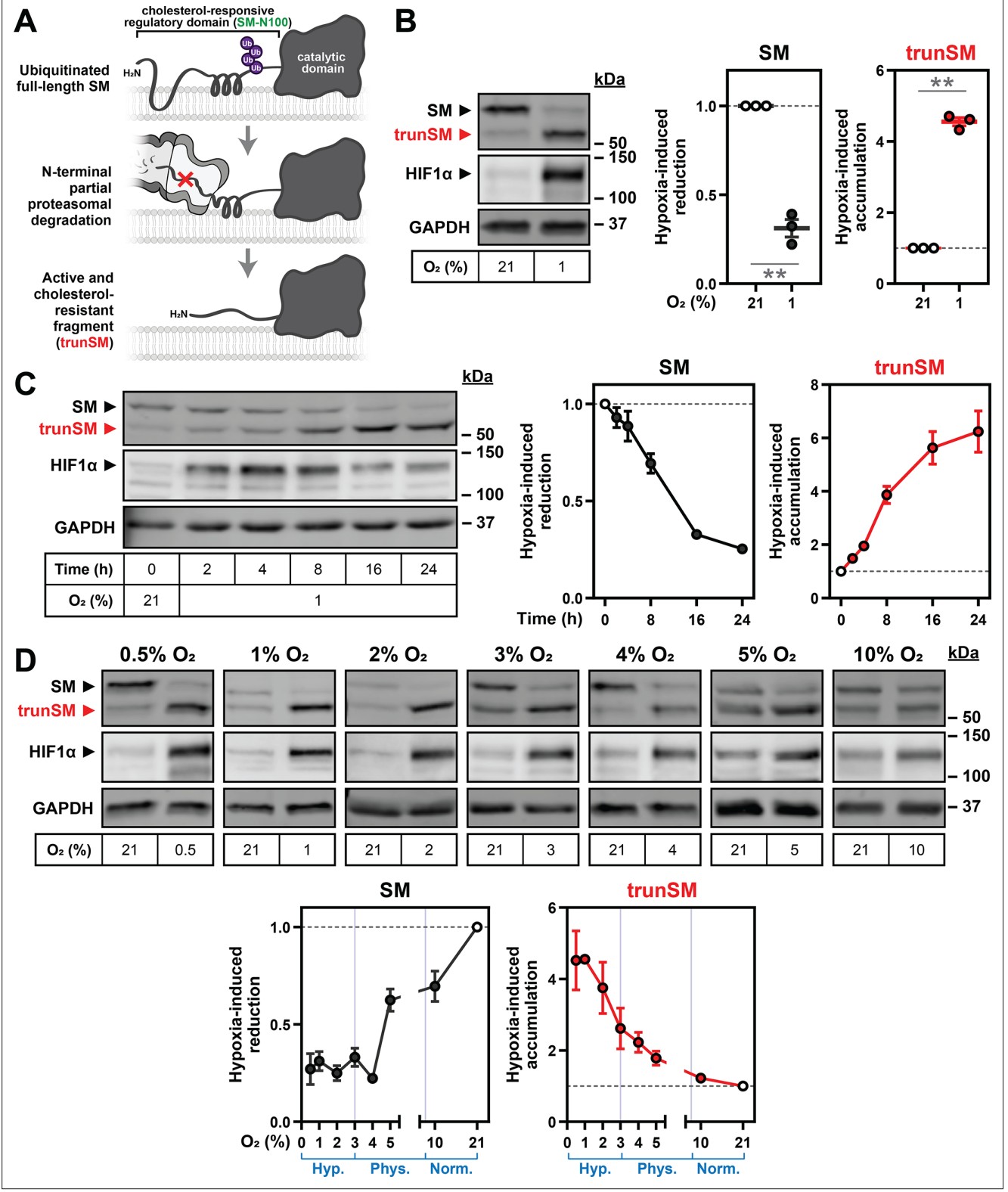

**Figure 1.** Oxygen availability regulates SM truncation. (**A**) Simplified overview of SM truncation. Full-length SM contains an N-terminal domain mediating feedback regulation by cholesterol. Ubiquitinated SM is targeted to the proteasome, where proteolysis is prematurely halted within the regulatory domain. This liberates a truncated protein (trunSM) that no longer responds to cholesterol and is therefore constitutively active. (**B**) HEK293T cells were incubated under normoxic (21% $O_2$) or hypoxic (1% $O_2$) conditions for 24 hr. (**C**) HEK293T cells were incubated under normoxic or hypoxic conditions for the indicated times. Changes in HIF1α levels over time are consistent with other reports (*Jantsch et al., 2011*; *Bartoszewski et al.,*

*Figure 1 continued on next page*

*Figure 1 continued*

*2019*). (**D**) HEK293T cells were incubated under the indicated oxygen concentrations for 24 hr. Each set of immunoblots was obtained in a separate experiment. (**B–D**) Immunoblotting was performed for SM and trunSM (red). Graphs depict densitometric quantification of SM and trunSM protein levels normalized to the normoxic condition, which was set to 1 (dotted line). In (D), oxygen concentrations considered hypoxic (hyp.), 'physoxic' (phys.) or normoxic (norm.) (*McKeown, 2014*) are indicated in blue. Data presented as mean ± SEM from n=3–4 independent experiments (**, p≤0.01; two-tailed one-sample *t*-test vs. hypothetical mean of 1).

The online version of this article includes the following source data and figure supplement(s) for figure 1:

**Source data 1.** Uncropped immunoblots for *Figure 1*.

**Figure supplement 1.** Simplified schematic of the cholesterol synthesis pathway.

**Figure supplement 2.** Hypoxia-induced truncation of SM is generalizable.

**Figure supplement 2—source data 1.** Uncropped immunoblots for *Figure 1—figure supplement 2*.

**Figure supplement 3.** Hypoxia-induced truncation of SM is independent of hypoxia-inducible factors.

**Figure supplement 3—source data 1.** Uncropped immunoblots for *Figure 1—figure supplement 3*.

(2) inhibition of its complete proteolysis. To confirm the first step of this mechanism, we investigated the reason for the decline in full-length SM levels during hypoxia. *SQLE* transcripts were downregulated in hypoxic HEK293T cells, as were transcripts encoding the upstream cholesterol synthesis enzyme HMGCR (*Figure 2A*). Downregulation of *SQLE* transcripts was not observed in MDA-MB-231 cells (*Figure 2—figure supplement 1A*), accounting for the unchanged full-length SM levels in this cell line. Although the reduction in *SQLE* and *HMGCR* transcripts in HEK293T cells likely reflected a broad transcriptional suppression of cholesterol synthesis during hypoxia, as reported previously (*Dolt et al., 2007*; *Cao et al., 2014*), the magnitude of *SQLE* downregulation was unlikely to fully explain the large reduction in SM protein levels (*Figure 1B*). Moreover, levels of a constitutively expressed SM construct ([HA]$_3$-SM-V5) were markedly reduced during extended hypoxic incubations with no associated change in mRNA levels (*Figure 2B*, *Figure 2—figure supplement 1B*). We concluded that hypoxia reduces the levels of full-length SM through both transcriptional downregulation and accelerated post-translational degradation.

The basal and metabolically-regulated degradation of SM occurs through the ubiquitin-proteasome system and is mediated by the SM-N100 regulatory domain (*Chua et al., 2017*; *Yoshioka et al., 2020*). Therefore, we tested the effect of hypoxia on HEK293 cells stably expressing an SM-N100 fusion protein (SM-N100-GFP-V5). Like full-length SM, levels of SM-N100-GFP-V5 were reduced by hypoxic conditions (*Figure 2C*). Proteasomal inhibition using MG132 increased the levels of SM and SM-N100-GFP-V5 (*Figure 2—figure supplement 1C*) and blocked their hypoxia-induced degradation (*Figure 2C*), confirming this degradation occurs via the proteasome. We also noted that protein levels and hypoxia-induced accumulation of trunSM were ablated by MG132 (*Figure 2—figure supplement 1C, D*), consistent with this protein arising from partial proteasomal proteolysis of SM (*Coates et al., 2021*). Although hypoxia can trigger autophagy (*Bellot et al., 2009*), this did not play a role in SM degradation as inhibition of lysosomal acidification using bafilomycin A1 had no additive effect with MG132 (*Figure 2C*). To identify residues required for hypoxia-induced degradation of SM, we utilized protein constructs with mutations of previously identified ubiquitination sites. The magnitude of hypoxic degradation was blunted by disruption of Lys-82/90/100, a cluster of redundant ubiquitination sites previously found to promote truncation (*Coates et al., 2021*), but not by disruption of Lys-290 (*Figure 2D*; *Hornbeck et al., 2015*). Non-canonical cysteine, serine and threonine ubiquitination sites required for the cholesterol-induced degradation of SM (SM-N100 C/S/T) (*Chua et al., 2019*) also contributed to hypoxia-induced degradation, suggesting multiple ubiquitin signals are involved. Contrary to the expectation that loss of ubiquitination would stabilize SM, the mutation of Lys-82/90/100 reduced SM levels under normoxic conditions (*Figure 2—figure supplement 1E*). Therefore, this cluster of residues may be specifically involved in hypoxia-induced, rather than basal, degradation of SM.

## Hypoxia-induced degradation of full-length SM requires the E3 ubiquitin ligase MARCHF6

To investigate how hypoxia promotes SM ubiquitination, we considered the possible role of proline hydroxylation. This oxygen-dependent modification, catalyzed by prolyl hydroxylases, is required for

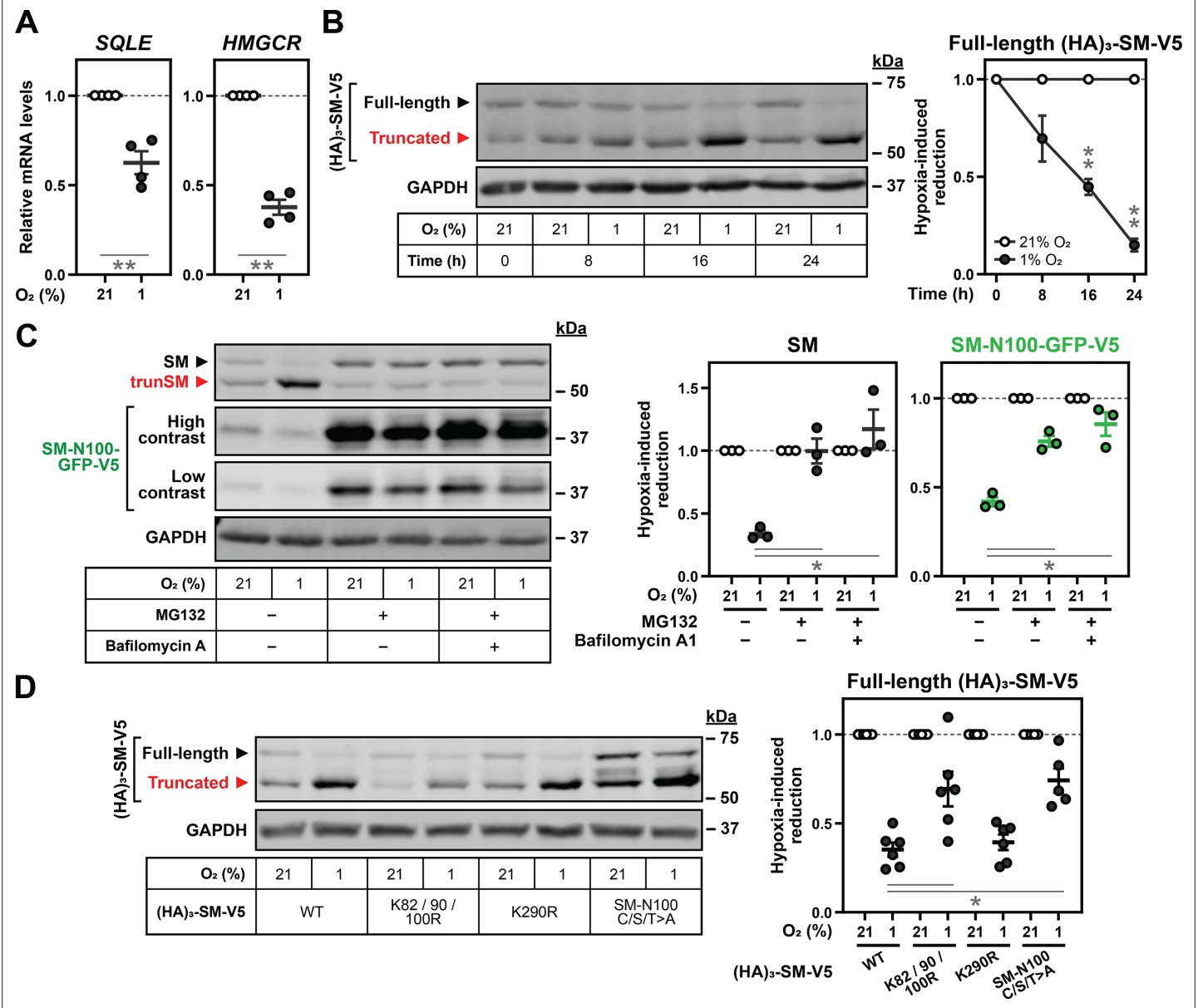

**Figure 2.** Hypoxia transcriptionally and post-translationally reduces full-length SM levels. (**A**) HEK293T cells were incubated under normoxic or hypoxic conditions for 24 hr. Levels of the indicated transcripts were quantified, normalized to the levels of *RPL11*, *GAPDH* and *ACTB* housekeeping transcripts and adjusted relative to the normoxic condition, which was set to 1 (dotted line). (**B**) HEK293T cells were transfected with (HA)$_3$-SM-V5 for 24 hr and incubated under normoxic or hypoxic conditions for the indicated times. (**C**) HEK SM-N100-GFP-V5 cells were treated with or without 20 μM MG132 and 20 nM bafilomycin A1 under normoxic or hypoxic conditions for 16 hr. (**D**) HEK293T cells were transfected with the indicated constructs for 24 hr and incubated under normoxic or hypoxic conditions for 16 hr. (**B–D**) Graphs depict densitometric quantification of protein levels normalized to the respective normoxic conditions for each timepoint, treatment or construct, which were set to 1 (dotted line). (**A–D**) Data presented as mean ± SEM from n=3–6 independent experiments (*, p≤0.05; **, p≤0.01; [A, B] two-tailed one-sample *t*-test vs. hypothetical mean of 1; [C, D] two-tailed ratio paired *t*-test).

The online version of this article includes the following source data and figure supplement(s) for figure 2:

**Source data 1.** Uncropped immunoblots for *Figure 2*.

**Figure supplement 1.** Hypoxia-induced degradation of full-length SM is via the ubiquitin-proteasome system.

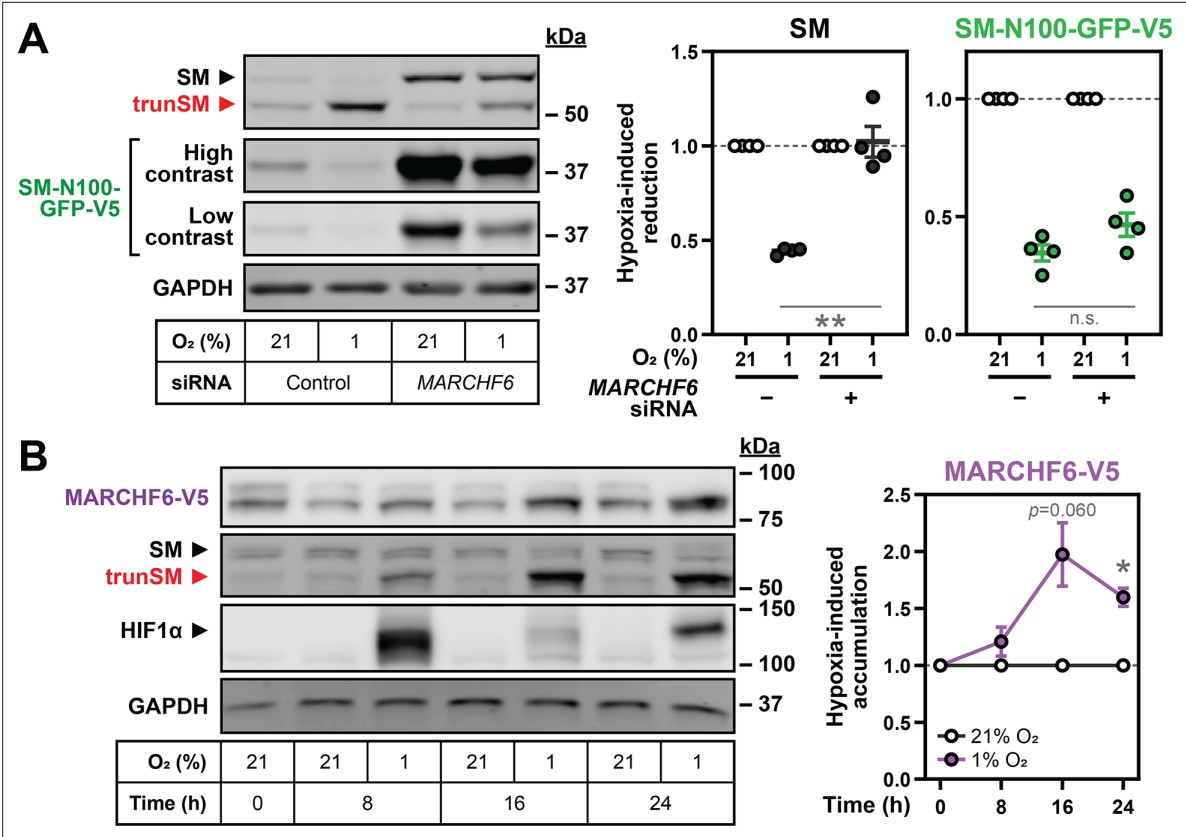

**Figure 3.** Hypoxia-induced degradation of full-length SM requires the E3 ubiquitin ligase MARCHF6. (**A**) HEK SM-N100-GFP-V5 cells were transfected with control or *MARCHF6* siRNA for 24 hr and incubated under normoxic or hypoxic conditions for 16 hr. (**B**) HEK MARCHF6-V5 cells were incubated under normoxic or hypoxic conditions for the indicated times. MARCHF6-V5 appears as two bands that were quantified collectively, as we have done previously (*Sharpe et al., 2019*). (**A, B**) Graphs depict densitometric quantification of protein levels normalized to the respective normoxic condition for each siRNA or timepoint, which were set to 1 (dotted line). Data presented as mean ± SEM from n=3–4 independent experiments (*, p≤0.05; **, p≤0.01; [A] two-tailed ratio paired *t*-test; [B] two-tailed one-sample *t*-test vs. hypothetical mean of 1).

The online version of this article includes the following source data and figure supplement(s) for figure 3:

**Source data 1.** Uncropped immunoblots for *Figure 3*.

**Figure supplement 1.** Hypoxia-induced degradation of full-length SM is independent of proline hydroxylation.

**Figure supplement 1—source data 1.** Uncropped immunoblots for *Figure 3—figure supplement 1*.

the ubiquitination and degradation of HIF1α under normoxic conditions (*Ivan et al., 2001*), although there is conflicting evidence for the existence of substrates beyond the HIF proteins (*Cockman et al., 2019*). Indeed, treatment with the prolyl hydroxylase inhibitors DMOG and FG-4592 had no effect on the basal levels nor hypoxia-induced degradation of SM and SM-N100-GFP-V5, despite stabilizing HIF1α (*Figure 3—figure supplement 1A*).

SM and SM-N100 are targeted for proteasomal degradation by the E3 ubiquitin ligase MARCHF6 (*Foresti et al., 2013*; *Zelcer et al., 2014*); therefore, we tested if increased MARCHF6 activity could account for the hypoxia-induced degradation of SM. To do so, we depleted *MARCHF6* expression using siRNA that achieves a 60–70% reduction in transcript levels in HEK293 cells (*Zelcer et al., 2014*). SM and SM-N100-GFP-V5 levels were dramatically increased (*Figure 3—figure supplement 1B*) and the hypoxic decline in SM levels was blocked (*Figure 3A*), supporting the involvement of MARCHF6 in hypoxia-induced degradation. The basal levels and hypoxic accumulation of trunSM were also reduced (*Figure 3—figure supplement 1B, C*), consistent with MARCHF6 contributing to the proteasomal targeting, and therefore partial degradation, of SM (*Coates et al., 2021*). Hypoxia-induced accumulation of trunSM was not completely abolished, however, indicating SM can be truncated even when targeted to the proteasome by hypoxia-independent mechanisms. Surprisingly, there was no effect of *MARCHF6* knockdown on hypoxia-induced degradation of SM-N100-GFP-V5 (*Figure 3A*), suggesting

SM and the isolated SM-N100 domain are degraded through different proteasome-dependent routes under these conditions. We elected to further investigate the MARCHF6-dependent degradation of full-length SM, as the endogenous protein has greater physiological relevance.

To study MARCHF6 levels in hypoxic cells, we used a previously generated HEK293 cell line stably expressing a V5-tagged form of the protein. This construct was used due to the poor performance of endogenous MARCHF6 antibodies (*Sharpe et al., 2019*) and to eliminate transcriptional effects on protein levels. We examined the response of MARCHF6-V5 to hypoxia and found it accumulated during prolonged hypoxic incubations, which correlated with the maximal decline in full-length SM levels (*Figure 3B*). Therefore, increased MARCHF6 protein levels and activity likely account for the accelerated ubiquitination and degradation of SM during hypoxia.

## Hypoxia-induced squalene accumulation promotes partial degradation of SM

Having established that SM undergoes accelerated proteasomal degradation during hypoxia, we next investigated how low oxygen levels favor its partial rather than complete proteolysis to yield trunSM. As there is extensive precedent for metabolic regulation of cholesterol synthesis enzymes and the pathway contains multiple oxygen-dependent reactions, we considered if accumulation of a pathway intermediate might underlie this phenomenon. Hypoxia-induced accumulation of trunSM occurred in cells incubated under both lipoprotein-replete and lipoprotein-deficient conditions, in which the cholesterol synthesis pathway is active (*Figure 4A*). However, the magnitude of this accumulation was diminished when lipoprotein-deficient cells were co-treated with a statin to inhibit HMGCR and the early cholesterol synthesis pathway. By contrast, there was no effect of sterol depletion on the hypoxia-induced reduction in full-length SM. This indicated that an intermediate or end-product of cholesterol synthesis promotes the partial rather than complete degradation of SM at the proteasome. We therefore turned our attention to the SM substrate squalene, as it allosterically regulates SM degradation (*Yoshioka et al., 2020*) and is the substrate for the first oxygen-dependent step of cholesterol synthesis. Squalene accumulated over the course of a hypoxic incubation (*Figure 4B*, *Figure 4—figure supplement 1*), consistent with reduced SM activity under low-oxygen conditions. This accumulation was strikingly well-correlated with the previously observed increase in trunSM levels (*Figure 4C*), suggesting the two effects may be linked.

Delivery of exogenous squalene induced trunSM accumulation in normoxic HEK293T and Huh7 cells (*Figure 4D*, *Figure 4—figure supplement 2A, B*), confirming its ability to promote partial degradation of SM. Accumulation of trunSM was also induced by the oxygenated squalene derivatives monooxidosqualene and dioxidosqualene (*Figure 4D*) but not by its saturated analogue squalane (*Figure 4—figure supplement 2C*), which has similar biophysical properties (*Hauß et al., 2002*). This indicated truncation is promoted by squalene and its structurally related molecules in a specific manner, rather than through bulk membrane effects caused by lipid accumulation. To address the possibility that exogenous squalene is converted to a downstream product responsible for truncation, we generated *SQLE*-knockout HEK293T cells (*Figure 4—figure supplement 3*) and transfected them with a catalytically inactive SM Y195F mutant (*Padyana et al., 2019*) to prevent the metabolism of added squalene. The truncated form of the Y195F mutant accumulated upon squalene treatment in *SQLE*-knockout cells, confirming squalene alone can directly induce truncation (*Figure 4—figure supplement 2D*). There was no significant accumulation of the truncated fragment in cells transfected with wild-type SM, likely due to clearance of exogenous squalene by the overexpressed protein and downstream enzymes.

To confirm if endogenously synthesized squalene is sufficient to trigger SM truncation, cells were treated with inhibitors of the relevant cholesterol synthesis enzymes (*Figure 4—figure supplement 2A*). The SM inhibitor NB-598 was excluded because of its ability to induce truncation through direct binding and stabilization of the SM catalytic domain that renders it resistant to proteasomal unfolding (*Padyana et al., 2019*; *Coates et al., 2021*). Inhibiting squalene synthesis from farnesyl diphosphate (TAK-475) increased trunSM levels but abolished its hypoxia-induced accumulation, whereas significant accumulation still occurred under conditions where squalene synthesis was preserved: inhibition of lanosterol synthesis from monooxidosqualene (BIBB 515), or inhibition of lanosterol demethylation (GR70585X) (*Figure 4—figure supplement 2E*). This confirmed that lanosterol, which also accumulates during hypoxia (*Nguyen et al., 2007*), has no effect on SM truncation. We further noted that

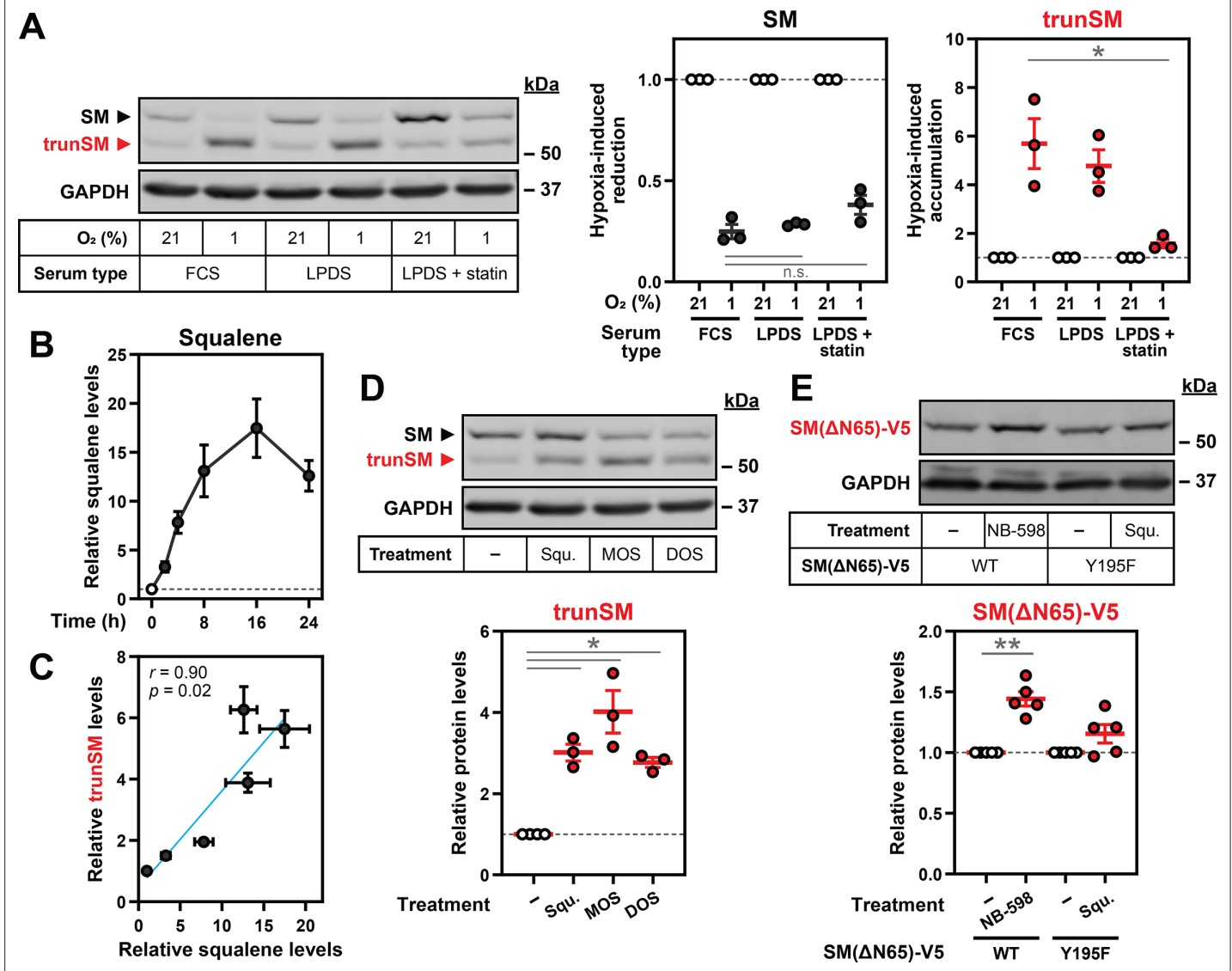

**Figure 4.** Hypoxia-induced squalene accumulation promotes partial degradation of SM. (**A**) HEK293T cells were incubated in medium containing fetal calf serum (FCS), lipoprotein-deficient FCS (LPDS) or LPDS containing 5 µM mevastatin and 50 µM mevalonolactone (LPDS +statin) for 8 hr, refreshed in their respective medium and incubated under normoxic or hypoxic conditions for 16 hr. (**B**) HEK293T cells were incubated under normoxic or hypoxic conditions for the indicated times. Non-saponifiable lipids were extracted, and squalene levels were determined using gas chromatography-mass spectrometry and adjusted relative to the normoxic condition, which was set to 1 (dotted line). The maximal squalene level detected was 0.66±0.12 ng per µg of total protein. (**C**) Pearson correlation between squalene levels in (B) and trunSM levels in *Figure 1C*. Blue line indicates linear regression. (**D**) HEK293T cells were treated with or without 300 µM squalene (squ.), monooxidosqualene (MOS) or dioxidosqualene (DOS) for 16 hr. (**E**) HEK293T *SQLE*-knockout (*SQLE*-KO) clone 10 (c10) cells were transfected with the indicated constructs for 24 hr, then treated with or without 1 µM NB-598 or 300 µM squalene for 16 hr. (**A, D, E**) Graphs depict densitometric quantification of trunSM or truncated protein levels normalized to the (A) respective normoxic conditions for each serum type or (D, E) vehicle conditions, which were set to 1 (dotted line). (**A–E**) Data presented as mean ± SEM from n=3–5 independent experiments (*, p≤0.05; **, p≤0.01; [A] two-tailed ratio paired *t*-test; [D, E] two-tailed one-sample *t*-test vs. hypothetical mean of 1).

The online version of this article includes the following source data and figure supplement(s) for figure 4:

**Source data 1.** Uncropped immunoblots for *Figure 4*.

**Figure supplement 1.** Hypoxia induces squalene accumulation.

**Figure supplement 2.** Farnesyl-containing cholesterol synthesis intermediates specifically promote partial degradation of SM.

**Figure supplement 2—source data 1.** Uncropped immunoblots for *Figure 4—figure supplement 2*.

**Figure supplement 3.** Generation of *SQLE*-knockout HEK293T cells.

**Figure supplement 3—source data 1.** Uncropped immunoblots for *Figure 4—figure supplement 3*.

*Figure 4 continued on next page*

*Figure 4 continued*

**Figure supplement 4.** Putative squalene-binding residues are not required for hypoxia-induced partial degradation of SM.

**Figure supplement 4—source data 1.** Uncropped immunoblots for *Figure 4—figure supplement 4*.

the inhibition of squalene or lanosterol synthesis, but not lanosterol demethylation, increased the levels of full-length SM and SM-N100-GFP-V5 under normoxic conditions. This was consistent with our previous finding that farnesyl-containing molecules, including monooxidosqualene, dioxidosqualene and a squalene-derived photoaffinity probe, stabilize SM via its regulatory domain in a similar manner to squalene itself (*Yoshioka et al., 2020*). The increase in normoxic trunSM levels upon treatment with TAK-475 and BIBB 515 (*Figure 4—figure supplement 2E*) suggested farnesyl-containing cholesterol synthesis intermediates can also induce SM truncation. Nevertheless, as the primary substrate of oxygen-dependent SM catalysis, squalene is likely to be the major driver of this process under hypoxic conditions.

SM contains two known squalene binding sites: the SM-N100 regulatory domain and the active site of the catalytic domain (*Yoshioka et al., 2020*). As the SM inhibitor NB-598 induces SM truncation by binding and stabilizing the catalytic domain (*Coates et al., 2021*), we considered if squalene exerts its effects on truncation through a similar mechanism. To eliminate the contribution of the SM-N100 domain, we transfected *SQLE*-knockout cells with an ectopic form of trunSM (SM[ΔN65]-V5). Consistent with past findings (*Yoshioka et al., 2020*; *Padyana et al., 2019*), this construct was stabilized by NB-598 (*Figure 4E*). However, significant stabilization did not occur when cells expressing an inactive SM(ΔN65)-V5 mutant (Y195F) were treated with squalene. We concluded that squalene promotes SM truncation via the SM-N100 regulatory domain, rather than direct binding and stabilization of the SM catalytic domain. Previous work showed squalene directly binds the SM-N100 domain (*Yoshioka et al., 2020*), with the hydrophobic re-entrant loop (residues ~ 15–40) as the most logical interaction site. Aromatic residues and leucine residues line the SM active site and are required for catalysis (*Padyana et al., 2019*; *Abe et al., 2007*), suggesting they directly interact with squalene. Similar residues in the SM-N100 re-entrant loop may likewise be involved in squalene binding, and possibly the partial degradation of SM. We therefore mutated phenylalanine and leucine residues in the re-entrant loop to test if they are required for SM truncation. However, these mutations, either alone or in combination, did not prevent the hypoxia-induced accumulation of truncated SM (*Figure 4—figure supplement 4*). This indicated that partial degradation of SM is independent of these putative squalene-binding residues, with further work required to clarify the mechanism.

## SM activity is preserved during hypoxia

trunSM has a long half-life and is constitutively active (*Coates et al., 2021*). Therefore, we hypothesized that hypoxia-induced truncation of SM is a compensatory mechanism to preserve enzymatic activity during oxygen shortfalls. To investigate this possibility, Huh7 cells were radiolabeled with [$^{14}$C]-acetate and downstream flux through cholesterol synthesis was assayed using thin-layer chromatography. This cell line was chosen due to its tissue of origin, the liver, being a major site of cholesterol synthesis (*Turley et al., 1995*), as well as its robust hypoxia-induced truncation of SM (*Figure 1—figure supplement 2F*). Cholesterol synthesis was markedly reduced in hypoxic cells (*Figure 5*), as expected given the high oxygen demand of the pathway. Acute lanosterol accumulation was detectable within 6 hr, consistent with previous reports (*Nguyen et al., 2007*; *Kucharzewska et al., 2015*), and continued over the duration of the hypoxic incubation. By contrast, upstream accumulation of squalene was minimal even after 24 hr. This confirmed that despite its dependence on oxygen, SM activity is preserved during hypoxia, likely due to its truncation to a degradation-resistant form.

## Discussion

Cholesterol synthesis is tightly regulated by metabolic supply and demand, with the lipid-sensing and rate-limiting enzyme SM a key point at which this regulation is exerted. We previously showed that partial proteasomal degradation of SM produces a truncated and constitutively active form of the enzyme (*Coates et al., 2021*). In this study, we identified hypoxia as a physiological trigger for trunSM formation (*Figure 1*). Hypoxia-induced truncation occurs through a two-part mechanism: (1) increased levels of MARCHF6, an E3 ubiquitin ligase that targets SM to the proteasome (*Figure 2*,

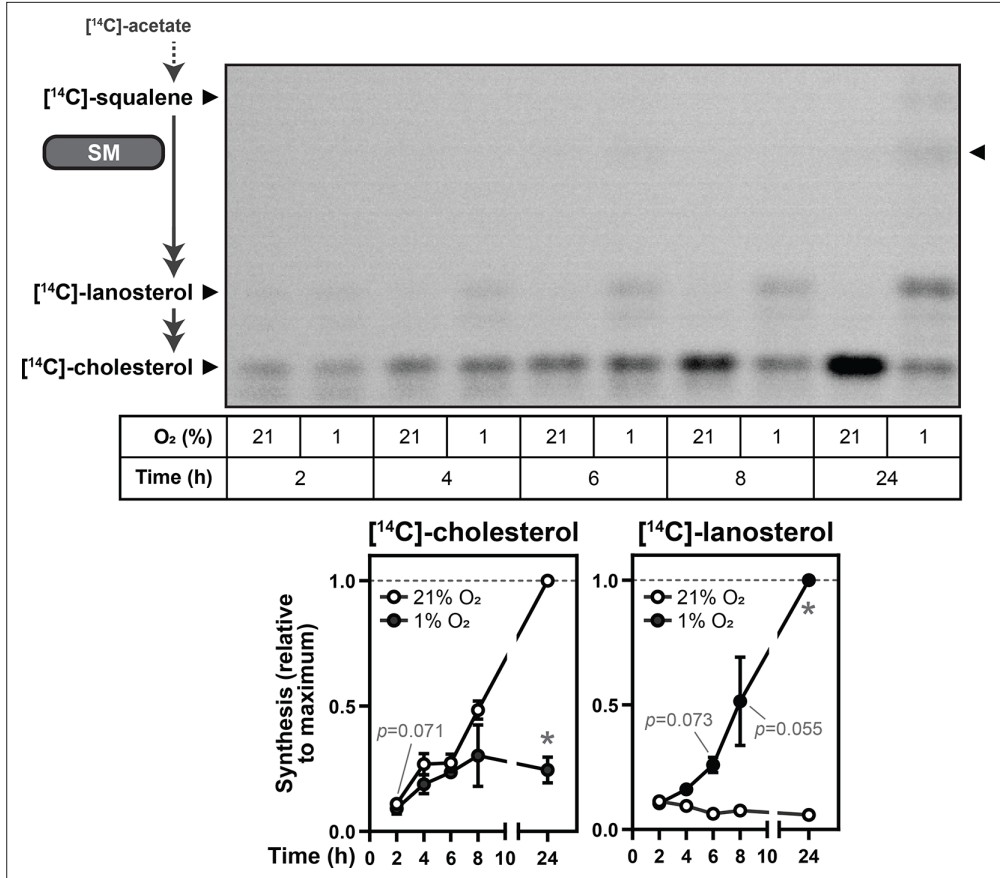

**Figure 5.** SM activity is preserved during hypoxia. Huh7 cells were labelled with 2 μCi [14C]-acetate and incubated under normoxic or hypoxic conditions for the indicated times. Synthesis of [14C]-squalene, [14C]-lanosterol, and [14C]-cholesterol was determined using thin-layer chromatography. Double-headed arrows indicate multiple enzymatic steps, and arrowhead on right of image indicates band corresponding to the SM product [14C]-monooxidosqualene (*Capell-Hattam et al., 2022*). Graphs depict densitometric quantification of [14C]-cholesterol and [14C]-lanosterol levels normalized to the conditions with the greatest abundance of each analyte, which were set to 1 (dotted line). Data presented as mean ± SEM from n=3 independent experiments (*, p≤0.05; two-tailed ratio paired *t*-test vs. respective normoxic condition for each timepoint).

The online version of this article includes the following source data for figure 5:

**Source data 1.** Uncropped thin-layer chromatography image for *Figure 5*.

*Figure 3*), and (2) accumulation of squalene, which impedes the complete degradation of SM and yields trunSM (*Figure 4*). Truncation of SM preserves its activity and facilitates downstream pathway flux during hypoxia (*Figure 5*). Taken together, our results point towards SM truncation as an adaptive mechanism to clear excess substrate, as well as a likely contributor to the widely reported oncogenic properties of SM.

## Cholesterol synthesis during hypoxia

Hypoxia places great strain on metabolic processes and necessitates the strict allotment of available oxygen and energy reserves. Cholesterol synthesis is a particularly resource-intensive pathway requiring eleven oxygen molecules and over one hundred ATP equivalents per molecule of product. However, there are conflicting reports on changes in overall flux from acetyl-CoA to cholesterol during hypoxia (*Mukodani et al., 1990*; *Parathath et al., 2011*), suggestive of cell type-specific responses. The small number of studies into individual biosynthetic enzymes nevertheless indicate their activity is suppressed by hypoxia at multiple regulatory levels, and this is supported by the accumulation of various pathway intermediates (*Wu et al., 2020*; *Nguyen et al., 2007*; *Kucharzewska et al., 2015*). Lanosterol 14α-demethylase, which requires three oxygen molecules for catalysis, is transcriptionally

downregulated by HIF2α and the hypoxia-induced long non-coding RNA *lincNORS*, contributing to the characteristic accumulation of lanosterol under hypoxic conditions (*Wu et al., 2020*; *Zhu et al., 2014*). Lanosterol in turn triggers ubiquitin-dependent degradation of the early cholesterol synthesis enzyme HMGCR (*Nguyen et al., 2007*), suppressing further oxygen consumption by the pathway. Our study establishes that oxygen availability also regulates SM, a rate-limiting enzyme of cholesterol synthesis and the first to require molecular oxygen.

We found that SM is transcriptionally downregulated in hypoxic HEK293T cells but not MDA-MB-231 cells, consistent with previously reported cell-type specific changes in *SQLE* expression (*Haider et al., 2016*). This accounted in part for the hypoxia-induced decline in full-length SM levels, although reduced *SQLE* translation through mechanisms such as mTOR suppression (*Liu et al., 2006*) cannot be ruled out as a contributing factor. We also found that like HMGCR, the SM protein is targeted for

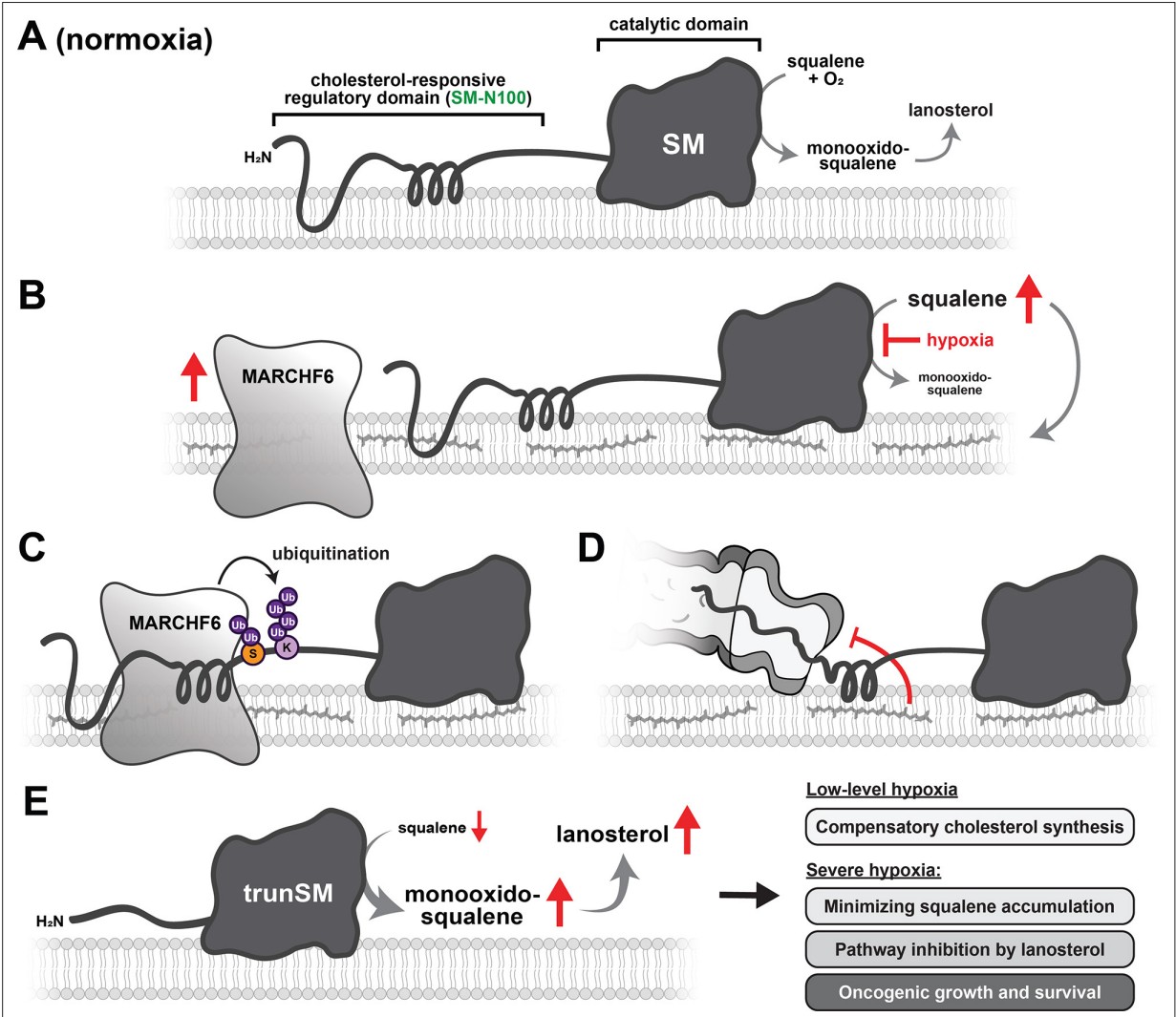

**Figure 6.** Model of hypoxia-induced SM truncation. (**A, B**) Hypoxic conditions stabilize the E3 ubiquitin ligase MARCHF6 and inhibit SM-catalyzed conversion of squalene to monooxidosqualene, leading to squalene accumulation. (**C**) Increased MARCHF6 activity promotes ubiquitination of SM and its targeting to the proteasome. (**D**) Squalene impedes the complete proteasomal degradation of SM via a mechanism involving the SM-N100 domain, (**E**) yielding the constitutively active trunSM. During transient or low-level hypoxia, trunSM activity may facilitate continued cholesterol synthesis to compensate for the oxygen shortfall. During long-term or severe hypoxia, trunSM activity may reduce squalene-induced toxicity and promote downstream synthesis of lanosterol, which suppresses an early step of the cholesterol synthesis pathway. During pathophysiological hypoxia, cholesterol synthesis enabled by trunSM may contribute to oncogenic cell growth and survival.

The online version of this article includes the following figure supplement(s) for figure 6:

**Figure supplement 1.** Summary of metabolic regulation of SM.

proteasomal degradation during hypoxia. However, in stark contrast to HMGCR, the net result of SM degradation under these conditions is the preservation of, or even an increase in, the number of enzyme molecules available for catalysis. Indeed, SM activity is preserved during hypoxia, with only minimal accumulation of its substrate squalene. This is enabled by increased partial proteolysis of SM to form trunSM, which lacks a functional SM-N100 regulatory domain and has a dramatically extended half-life (*Coates et al., 2021*). Disruption of the SM-N100 domain also renders trunSM resistant to cholesterol-induced degradation, which is central to the metabolic regulation of full-length SM (*Gill et al., 2011*; *Coates et al., 2021*). Therefore, hypoxia-induced SM truncation ensures total enzyme levels remain constant even under cholesterol-replete conditions that typically trigger its degradation.

Although preservation of oxygen-dependent SM activity during hypoxia appears paradoxical, there are numerous advantages (*Figure 6*). During transient or low-level hypoxia, it likely promotes compensatory cholesterol synthesis and maintenance of cell viability. Supporting this idea, hypoxia-induced cell death is exacerbated by SM inhibition (*Haider et al., 2016*). Furthermore, the longevity of trunSM would enable rapid resumption of pathway activity when normoxia is restored. During long-term or severe hypoxia, where there is insufficient oxygen for flux through downstream cholesterol synthesis, the role of trunSM in hypoxia may shift towards efficient clearance of squalene. While generally considered an inert intermediate, excess squalene induces ER stress and is toxic in cells lacking SM activity or the ability to sequester squalene to lipid droplets (*Hong et al., 2022*; *Mahoney et al., 2019*). A secondary effect of squalene clearance is its downstream conversion to lanosterol, which accelerates degradation of HMGCR (*Nguyen et al., 2007*). We and others (*Nguyen et al., 2007*; *Kucharzewska et al., 2015*) observed a stark contrast between acute lanosterol accumulation and minimal squalene accumulation during hypoxia, supporting both of these putative functions for trunSM. Thus, hypoxia-induced SM truncation is likely critical for curtailing HMGCR activity and flux through oxygen-intensive cholesterol synthesis. Stabilization of MARCHF6, which targets lanosterol 14α-demethylase for degradation (*Scott et al., 2020*), may also contribute to hypoxic lanosterol accumulation. SM activity also promotes synthesis of dioxidosqualene and ultimately 24(S),25-epoxycholesterol, a potent suppressor of cholesterol accretion (*Wong et al., 2008*). However, these are yet to be studied in the context of hypoxia.

## Molecular mechanism of truncation

The first step of SM truncation is its delivery to the proteasome (*Coates et al., 2021*). This is enhanced during hypoxia by a novel ubiquitin signal at the Lys-82/90/100 cluster and accumulation of MARCHF6, which facilitates the basal and metabolically-regulated degradation of SM (*Foresti et al., 2013*; *Zelcer et al., 2014*). Non-canonical ubiquitination sites required for cholesterol-induced SM degradation (*Chua et al., 2019*) also contribute to its hypoxic degradation, which can be reconciled by a combination of increased MARCHF6 levels and the sterol-replete conditions in which these experiments were performed. The mechanism by which hypoxia stabilizes MARCHF6 requires further study. Interestingly, NADPH binds MARCHF6 and activates its E3 ubiquitin ligase activity (*Nguyen et al., 2022*). Hypoxic cells upregulate NADPH production to counter oxidative stress (*Samanta et al., 2016*), which may contribute to MARCHF6-dependent proteasomal targeting of SM under these conditions.

A key finding of this study is that upon hypoxic delivery of SM to the proteasome, accumulated squalene inhibits its complete degradation and liberates the constitutively active trunSM. This feed-forward mechanism, by which the substrate of SM preserves its activity, is mediated by the SM-N100 regulatory domain. It also functions under normoxic conditions, where it likely buffers against transient squalene fluctuations. Hypoxia-induced trunSM accumulation persists even when MARCHF6 is depleted; therefore, squalene also impedes SM degradation when other E3 ubiquitin ligases target it to the proteasome. These other regulators, and the factors controlling their ubiquitination of SM, await future discovery and may shed light on how the SM-N100 domain undergoes MARCHF6-independent proteasomal degradation during hypoxia. Taken together, our findings expand on the ability of the SM-N100 domain to sense ER membrane lipids and regulate SM degradation (*Figure 6—figure supplement 1*).

Although hypoxia and lipid accumulation trigger ER stress and changes to proteostasis (*Bi et al., 2005*), the ability of squalene and other farnesyl-containing compounds to allosterically bind the SM-N100 domain (*Yoshioka et al., 2020*) strongly suggests truncation is induced directly by squalene, rather than bulk membrane effects. The specificity of this effect is supported by the inability of the saturated squalene analog, squalane, to induce truncation. Squalene is the most likely of the farnesyl-containing cholesterol synthesis intermediates to accumulate under both normal and hypoxic conditions, owing to the rate-limiting and oxygen-dependent nature of SM activity. However, our current and previous (*Yoshioka et al., 2020*) data suggest a farnesyl group alone is sufficient to promote truncation. Precisely how this lipophilic moiety interferes with SM degradation is unknown, although interaction with the membrane-embedded SM-N100 re-entrant loop (*Howe et al., 2015*) is a likely possibility. We ruled out putative squalene-binding residues within this loop as a requirement for hypoxia-induced truncation, although a stronger understanding of the exact squalene binding site in SM-N100 is needed to further test its role in SM truncation.

Previously, we reported farnesyl-containing compounds also blunt the MARCHF6-mediated ubiquitination of SM-N100, leading to stabilization of full-length SM (*Yoshioka et al., 2020*). The existence of dual mechanisms enabling squalene to sustain SM activity is intriguing, particularly as trunSM formation is itself dependent on ubiquitination (*Coates et al., 2021*). One possibility is that truncation is stimulated at a lower squalene threshold than the inhibition of SM ubiquitination, allowing for a biphasic response to accumulating substrate. We found squalene is at the lower limit of detection in normoxic cells, and only the highly sensitive gas chromatography-mass spectrometry method could detect its accumulation during acute hypoxia. The strong trunSM-squalene correlation thus suggests truncation is triggered by very small quantities of squalene. Another possibility is that squalene-induced truncation is a 'failsafe' mechanism to preserve SM activity. This may be necessary when a reduction in MARCHF6-mediated ubiquitination is insufficient to completely prevent targeting of SM to the proteasome. It may also apply when SM ubiquitination is promoted by other, simultaneous cellular stimuli. For instance, hypoxic MARCHF6 stabilization may override the inhibitory effects of accumulated squalene.

## Physiological consequences of truncation

Overexpression and overactivity of SM occurs in a wide range of malignancies including hepatocellular carcinoma and prostate cancer, where it is positively correlated with severity and lethality (*Liu et al., 2018*; *Kalogirou et al., 2021*). As hypoxia is common in the interior of solid tumors and often associated with poor prognosis (*Rankin and Giaccia, 2016*), trunSM formation may contribute to oncogenesis. Hypoxia also occurs within atherosclerotic plaques (*Marsch et al., 2013*). Truncation of SM is yet to be specifically studied in human tissues, and would be worthwhile to test in cancers with a propensity for both hypoxia and SM overexpression, such as prostate and pancreatic cancer (*McKeown, 2014*). trunSM retains full catalytic activity (*Coates et al., 2021*) and can likely fulfil cholesterol synthesis-dependent functions of SM in oncogenesis. However, its competency in the suite of other oncogenic SM activities is unknown. These include activation of ERK signaling (*He et al., 2021*) and interactions with carbonic anhydrase 3 (*Liu et al., 2021*), GSK3β and p53 (*Jun et al., 2021*). Cholesterol-independent functions such as these are likely to be most critical during severe hypoxia, where cholesterol synthesis cannot proceed.

Finally, although this study was largely performed using extremely low oxygen levels characteristic of pathophysiological hypoxia, we also observed that SM truncation was responsive to a range of 'physoxic' oxygen levels that occur within normal tissues in situ. This reinforces the concept of truncation as a buffer against normal substrate fluctuations, which may be important for preventing squalene-induced dysfunction in tissues with low oxygen perfusion, such as the brain (*McKeown, 2014*). Squalene fluctuations occur diurnally (*Miettinen, 1982*) and with changing sterol status (*Gill et al., 2011*), and its levels vary dramatically between different tissues (*Liu et al., 1976*). Thus, the ability of squalene to stimulate the truncation and constitutive activation of SM may play a key role in regulating flux through cholesterol synthesis under various physiological conditions.

# Materials and methods

## Key resources table

| Reagent type (species) or resource | Designation | Source or reference | Identifiers | Additional information |
|---|---|---|---|---|
| Gene (*Homo sapiens*) | *SQLE* | RefSeq | 6713, NM_003129.4 | |
| Cell line (*H. sapiens*) | HEK293T | Gift from UNSW School of Medical Sciences (UNSW Sydney, Australia) | | Highly transfectable human embryonic kidney cells |
| Cell line (*H. sapiens*) | HCT116 | Gift from Dr Ewa Goldys (UNSW Sydney, Australia) | | Epithelial colorectal carcinoma cells |
| Cell line (*H. sapiens*) | Huh7 | Gift from Centre for Cardiovascular Research (UNSW Sydney, Australia) | | Epithelial-like hepatocellular carcinoma cells |
| Cell line (*H. sapiens*) | HeLaT | Gift from Drs Louise Lutze-Mann and Noel Whitaker (UNSW Sydney, Australia) | | Highly transfectable cervical adenocarcinoma cells |
| Cell line (*H. sapiens*) | MDA-MB-231 | Gift from Drs Louise Lutze-Mann and Noel Whitaker (UNSW Sydney, Australia) | | Epithelial breast adenocarcinoma cells |
| Cell line (*H. sapiens*) | HEK SM-N100-GFP-V5 | Generated previously using the Flp-In T-REx system (**Zelcer et al., 2014**) | | HEK293 cells stably expressing the N-terminal 100 amino acids of SM (SM-N100) fused with green fluorescent protein and a V5 epitope tag. Controlled by a cytomegalovirus (CMV) promoter. |
| Cell line (*H. sapiens*) | HEK MARCHF6-V5 | Generated previously using the Flp-In T-REx system (**Sharpe et al., 2019**) | | HEK293 cells stably expressing MARCHF6 fused with a V5 epitope tag. Controlled by a CMV promoter. |
| Cell line (*H. sapiens*) | HEK293T SQLE-knockout clone 10 (c10) | This study | | See Materials and Methods, Generation of SM knockout cells |
| Cell line (*H. sapiens*) | HEK293T SQLE-knockout clone 12 (c12) | This study | | See Materials and Methods, Generation of SM knockout cells |
| Cell line (*H. sapiens*) | HEK293T SQLE-knockout clone 14 (c14) | This study | | See Materials and Methods, Generation of SM knockout cells |
| Transfected construct (*H. sapiens*) | MISSION universal negative control #1 siRNA | Sigma-Aldrich | SIC001 | |
| Transfected construct (*H. sapiens*) | *MARCHF6* siRNA | Sigma-Aldrich | SASI_Hs01_00105239 | |
| Transfected construct (*H. sapiens*) | *HIF1A* siRNA | Sigma-Aldrich | SASI_Hs01_00332063 | |
| Transfected construct (*H. sapiens*) | *EPAS1* (*HIF2A*) siRNA | Sigma-Aldrich | SASI_Hs01_00019152 | |
| Antibody | Anti-SM(SQLE) (rabbit polyclonal) | Proteintech | 12544-1-AP | 4 °C for 16 hr (1:2500) |
| Antibody | Anti-HIF1α (rabbit polyclonal) | Proteintech | 20960-1-AP | Room temperature for 1 hr (1:1000) |
| Antibody | Anti-GAPDH (rabbit monoclonal) | Cell Signaling Technology | 2118 | 4 °C for 16 hr (1:2000) |

*Continued on next page*

*Continued*

| Reagent type (species) or resource | Designation | Source or reference | Identifiers | Additional information |
|---|---|---|---|---|
| Antibody | Anti-V5 (mouse monoclonal) | Thermo Fisher Scientific | R960-25 | Room temperature for 1 hr (1:5000) |
| Antibody | IRDye 680RD anti-rabbit IgG (donkey polyclonal) | LI-COR Biosciences | LCR-926-68073 | Room temperature for 1 hr (1:5000) |
| Antibody | IRDye 800CW anti-mouse IgG (donkey polyclonal) | LI-COR Biosciences | LCR-926-32212 | Room temperature for 1 hr (1:10,000) |
| Antibody | Peroxidase-conjugated AffiniPure anti-rabbit IgG (donkey polyclonal) | Jackson ImmunoResearch Laboratories | 711-035-152 | Room temperature for 1 hr (1:10,000) |
| Antibody | Peroxidase-conjugated AffiniPure anti-mouse IgG (donkey polyclonal) | Jackson ImmunoResearch Laboratories | 715-035-150 | Room temperature for 1 hr (1:10,000) |
| Commercial assay or kit | Lipofectamine RNAiMAX transfection reagent | Thermo Fisher Scientific | 13778150 | |
| Commercial assay or kit | Lipofectamine 3000 transfection reagent | Thermo Fisher Scientific | L3000001 | |
| Commercial assay or kit | Bicinchoninic acid assay kit | Thermo Fisher Scientific | 23225 | |
| Commercial assay or kit | TRI reagent | Thermo Fisher Scientific | AM9738 | |
| Commercial assay or kit | SuperScript III First-Strand Synthesis kit | Thermo Fisher Scientific | 18080051 | |
| Commercial assay or kit | PureLink Genomic DNA Mini kit | Thermo Fisher Scientific | K182001 | |
| Commercial assay or kit | QuantiNova SYBR Green PCR kit | Qiagen | 208052 | |
| Commercial assay or kit | Immobilon Western chemiluminescent HRP substrate | Millipore | WBKLS0500 | |
| Chemical compound, drug | [$^{14}$C]-Acetic acid sodium salt | PerkinElmer | NEC084H001MC | |
| Chemical compound, drug | 2,3,22,23-Dioxidosqualene | Echelon Biosciences | S-0302 | |
| Chemical compound, drug | 2,3-Oxidosqualene (mono-oxidosqualene) | Echelon Biosciences | S-0301 | |
| Chemical compound, drug | 5α-Cholestane | Sigma-Aldrich | C8003 | |
| Chemical compound, drug | Bafilomycin A1 | Sigma-Aldrich | B1793 | |
| Chemical compound, drug | BIBB 515 | Cayman Chemical Company | 10010517 | |
| Chemical compound, drug | DMOG | Sigma-Aldrich | D3695 | |

*Continued on next page*

*Continued*

| Reagent type (species) or resource | Designation | Source or reference | Identifiers | Additional information |
|---|---|---|---|---|
| Chemical compound, drug | FG-4592 | Cayman Chemical Company | 15294 | |
| Chemical compound, drug | GR70585X | GlaxoSmithKlein | N/A | |
| Chemical compound, drug | Mevalonolactone | Sigma-Aldrich | M4667 | |
| Chemical compound, drug | Mevastatin | Sigma-Aldrich | M2537 | |
| Chemical compound, drug | MG132 | Sigma-Aldrich | C2211 | |
| Chemical compound, drug | NB-598 | Chemscene | CS-1274 | |
| Chemical compound, drug | Squalane | Sigma-Aldrich | 234311 | |
| Chemical compound, drug | Squalene | Sigma-Aldrich | S3626 | |
| Chemical compound, drug | TAK-475 | Sigma-Aldrich | SML2168 | |
| Chemical compound, drug | *N,O*-Bis-(trimethylsilyl) trifluoroacetamide | Supelco | T6381 | |
| Software, algorithm | Thermo Xcalibur software | Thermo Fisher Scientific | v2.2 SP1.48 | |
| Software, algorithm | Image Studio Lite software | LI-COR Biosciences | v5.2.5 | |
| Software, algorithm | GraphPad Prism software | GraphPad Software Inc | v9.0 | |
| Other | Opti-MEM I reduced serum medium | Thermo Fisher Scientific | 31985062 | Used to deliver transfection complexes. See Materials and Methods, siRNA and Plasmid Transfection |

## Cell culture

The cell lines used in this study are listed in the Key Resources Table and were routinely tested to ensure they were mycoplasma-free. Cells were maintained in a humidified Heraeus BB 15 incubator at 37 °C, 5% $CO_2$, and 21% $O_2$ (normoxia) in maintenance medium (DMEM-HG, 10% [v/v] fetal calf serum [FCS], 100 U/mL penicillin, and 100 µg/mL streptomycin). To improve HEK293 and HEK293T cell surface adhesion, culture vessels were treated with 25 µg/mL polyethyleneimine in phosphate-buffered saline (PBS) for 15 min at 37 °C prior to cell seeding. Plasmid and siRNA transfections were performed in maintenance medium lacking penicillin and streptomycin. Sterol depletions were performed in maintenance medium containing lipoprotein-deficient serum (LPDS, 30 mg/mL protein), which was prepared from FCS by density gradient centrifugation and dialysis, as described in *Goldstein et al., 1983*. Hypoxic incubations at 0.5–10% $O_2$ were performed in a humidified Binder CB 150 incubator at 37 °C and 5% $CO_2$. For all treatments (listed in the Key Resources Table), appropriate solvent controls were used (water [DMOG]; ethanol (mevalonolactone, GR70585X); DMSO [MG132, bafilomycin A1, FG-4592, mevastatin, TAK-475, BIBB-515, NB-598]; dimethyl sulfoxide containing 1% [v/v] Tween-20 [squalene, monooxidosqualene, dioxidosqualene, squalane]) and the final concentration of solvent

did not exceed 1% (v/v) in cell culture medium. Treatments were delivered in full medium refreshes, and all experiments were 48–72 hr in duration.

## Plasmids

Plasmids encoding Cas9 and *SQLE*-targeting guide RNAs were generated by BbsI cloning into a PX458 vector as described in *Ran et al., 2013*. Amino acid substitutions within expression vectors were generated using the overlap extension cloning method, as described previously (*Stevenson et al., 2013*). The identity of all plasmids was confirmed via Sanger sequencing. The plasmids used in this study are listed in *Appendix 1—table 1*, and the primer sequences used for DNA cloning are listed in *Appendix 1—table 2*.

## siRNA and plasmid transfection

To downregulate gene expression or transiently overexpress proteins, cells were seeded into 12-well plates. The next day, cells were transfected with 15 pmol siRNA using Lipofectamine RNAiMAX (15 pmol siRNA: 2 μL reagent) or 1 μg expression vector using Lipofectamine 3000 (1 μg DNA: 2 μL reagent with 2 μL P3000 supplemental reagent), delivered in Opti-MEM I reduced serum medium. After 24 hr, cells were refreshed in maintenance medium and treated as specified in figure legends. The siRNA used in this study are listed in the Key Resources Table.

## Protein harvest and immunoblotting

To quantify protein levels, cells were seeded into 6- or 12-well plates and treated as specified in figure legends. For detection of SM, total protein was harvested in 2% SDS lysis buffer (10 mM Tris-HCl [pH 7.6], 2% [w/v] SDS, 100 mM sodium chloride, 2% [v/v] protease inhibitor cocktail), passed through a 21-gauge needle until homogenous, and vortexed at room temperature for 20 min. For detection of MARCHF6-V5, cells were scraped in ice-cold PBS, pelleted, and lysed in modified RIPA buffer (50 mM Tris-HCl [pH 8.0], 0.1% [w/v] SDS, 1.5% [w/v] IGEPAL CA-630, 0.5% [w/v] sodium deoxycholate, 150 mM sodium chloride, 2 mM magnesium chloride, 2% [v/v] protease inhibitor cocktail), passed 20 times through a 22-gauge needle, rotated at 4 °C for 30 min, and centrifuged at 17,000 $g$ and 4 °C for 15 min to obtain the supernatant. Lysate protein content was quantified using the bicinchoninic acid (BCA) assay, and sample concentrations were normalized by dilution in the appropriate lysis buffer and 0.25 vol 5×Laemmli buffer (250 mM Tris-HCl [pH 6.8], 10% [w/v] SDS, 25% [v/v] glycerol, 0.2% [w/v] bromophenol blue, 5% [v/v] β-mercaptoethanol). For SM detection, normalized samples were heated at 95 °C for 5 min.

Proteins were separated on 10% (w/v) Tris-glycine SDS-PAGE gels (prepared in-house), electroblotted onto nitrocellulose membranes, and blocked in 5% (w/v) skim milk powder in PBS containing 0.1% (v/v) Tween-20 (PBST). Immunoblotting was performed using the antibodies listed in the Key Resources Table, which were diluted in 5% (w/v) bovine serum albumin in PBST containing 0.02% (w/v) sodium azide, except for anti-HIF1α and peroxidase-conjugated antibodies, which were diluted in 5% (w/v) skim milk powder in PBST. Fluorescence-based detection of SM, trunSM, GAPDH, SM-N100-GFP-V5, and (HA)$_3$-SM-V5 was performed using an Odyssey CLx imager (LI-COR Biosciences), and enhanced chemiluminescence-based detection of HIF1α and MARCHF6-V5 was performed using Immobilon Western chemiluminescent HRP substrate (Millipore) and an ImageQuant LAS 500 imager (Cytiva Life Sciences). Densitometry analysis was performed using Image Studio Lite software.

## RNA harvest and qRT-PCR

To quantify gene expression, cells were seeded into 12-well plates and treated as specified in figure legends. Total RNA was harvested using TRI reagent and polyadenylated RNA was reverse transcribed using the SuperScript III First Strand Synthesis kit. cDNA products were used as the template for quantitative reverse transcription-PCR (qRT-PCR) in technical triplicate using the QuantiNova SYBR Green PCR kit and primers listed in *Appendix 1—table 2*. mRNA levels were normalized to the geometric mean of *RPL11*, *GAPDH,* and *ACTB* for hypoxia experiments, or *PBGD* for validation of siRNA knockdowns and gene knockout, using the comparative $C_T$ method (*Schmittgen and Livak, 2008*). Normalized data were adjusted relative to the control condition, as specified in figure legends.

## Gas chromatography-mass spectrometry

To quantify cellular squalene levels, gas chromatography-mass spectrometry was performed as described previously (*Yoshioka et al., 2020*). Briefly, cells in 6-well plates were lysed in 0.05 M sodium

hydroxide, total protein was quantified using the BCA assay, and samples were adjusted to the lowest protein concentration using 0.05 M sodium hydroxide plus 4 µg 5α-cholestane as an internal standard, in a total volume of 1 mL. Lysates were mixed with 1 mL 100% (v/v) ethanol, 500 µL 75% (w/v) potassium hydroxide, 1 µL 20 mM butylated hydroxytoluene, and 20 µL 20 mM EDTA, and saponified at 70 °C for 1 hr. Non-saponifiable lipids were extracted by mixing with 1 mL 100% (v/v) ethanol and 2.5 mL hexane, centrifuging at 4,000 $g$ for 5 min, and collection of 2 mL of the organic phase. Lipids were dried in a vacuum centrifuge, resuspended in 50 µL $N,O$-bis(trimethylsilyl)-trifluoroacetamide, and derivatized at 60 °C for 1 hr.

Derivatized lipids (1.5 µL) were injected via a heated (300 °C) splitless with surge (38.0 psi for 0.50 min) inlet into a Thermo Trace gas chromatograph fitted with a Trace TR-50MS GC column (60 m×0.25 mm, 0.25 µm film thickness, Thermo Fisher). Analytes were separated with helium as the carrier gas at a constant flow of 1.2 ml/min with vacuum compensation, and temperature programming as follows: 70 °C for 0.70 min, 20 °C/min to 250 °C, 3 °C/min to 270 °C, 1.5 °C/min to 315 °C, then hold for 10 min. The GC column was coupled to a Thermo DSQIII mass spectrometer, with a transfer line temperature of 320 °C and an ion source temperature of 250 °C. For mass spectrometry analysis, the electron energy was 70 eV, the emission current was 130 µA and the detector gain was $3.0×10^5$. Squalene and 5α-cholestane standards were analyzed in scan mode (34–600 Da) to identify peaks and retention times, and identity was confirmed using the National Institute of Standards and Technology databases. Experimental samples were analyzed in selective ion monitoring mode to detect squalene ($m/z$=81.0, 410.4) and 5α-cholestane ($m/z$=149.1, 217.2, 372.4), with a detection width of 0.1 and dwell time of 200 ms. Chromatograph peaks were integrated using Thermo Xcalibur software and the peak area of squalene was normalized to the 5α-cholestane internal standard. A squalene standard curve ranging from 3.125–100 ng/µL was used to quantify squalene levels, with data adjusted to the total protein content of the cell lysate.

## Generation of SM knockout cells

To knock out SM using CRISPR/Cas9, guide RNAs targeting the *SQLE* proximal promoter (hg38 chr8:124998048–124998067; AATGGAAACGTTCCGACCCG) and first exon (hg38 chr8:124999588–124999607; ATCCGAGAAGAGGGCGAACT) were designed using CHOPCHOP (*Labun et al., 2019*) and cloned into the Cas9- and GFP-encoding PX598 vector using BbsI restriction enzyme cloning, as described in *Ran et al., 2013*, and primers listed in *Appendix 1—table 2*. HEK293T cells were seeded into 10 cm dishes and transfected with 5 µg of both vectors using Lipofectamine 3000, as described above. After 24 hr, cells were trypsinized, washed with PBS and resuspended in FACS buffer (5% [v/v] FCS and 10 mM EDTA in PBS). Cells were sorted based on GFP fluorescence using a BD FACSAria III at the UNSW Flow Cytometry Facility. GFP-positive cells were seeded into 10 cm dishes at a low density (6,000–12,000 cells/dish) and allowed to adhere for 2–3 weeks. Single colonies were picked, expanded, and screened for *SQLE* mRNA expression and SM protein expression via qRT-PCR and immunoblotting, as described above. To identify the genomic lesion, genomic DNA was isolated from clones using the PureLink Genomic DNA Mini kit. The CRISPR/Cas9 target region was amplified and cloned into a pcDNA3.1 vector for Sanger sequencing.

## Cholesterol synthesis assays

To assay flux through cholesterol synthesis, radiolabeling and thin-layer chromatography were performed as described previously (*Capell-Hattam et al., 2022*). Briefly, cells were seeded into 6-well plates and incubated in maintenance medium containing 2 µCi/well [$^{14}$C]-acetic acid sodium salt as specified in figure legends. Cells were lysed in 1 mL 0.05 M sodium hydroxide and protein content was determined using the bicinchoninic acid assay. Samples were normalized to the lowest protein concentration within each experiment by discarding the required volume and making up to 1 mL with 0.05 M sodium hydroxide. Lysates were mixed with 1 mL ethanol, 500 µL 75% (w/v) potassium hydroxide, 1 µL 20 mM butylated hydroxytoluene, and 20 µL 20 mM EDTA, and saponified by incubation in a 70 °C water bath for 1 hr. Once cooled to room temperature, non-saponifiable lipids were extracted by mixing samples with 1 mL ethanol and 2.5 mL hexane, vortexing for 30 s, and centrifuging at 2,000 $g$ for 5 min. The upper organic phase (2 mL) was collected and evaporated to dryness in a fume cupboard. Dried lipids were resuspended in 50 µL hexane and separated by thin layer chromatography on TLC Silica gel 60 F254 plates (Supelco) using a heptane:ethyl acetate (2:1, v/v)

mobile phase. Silica plates were dried, exposed to BAS-IP SR phosphor screens (Fujifilm) for 5–7 days, and imaged using a Typhoon FLA 9500 imager (GE Healthcare). Densitometry analysis was performed using Image Studio Lite software and levels of each analyte were normalized to the condition with the highest abundance.

## Data analysis and presentation

Data were normalized as described in figure legends. To quantify overall changes in the predominant SM variant, the trunSM:SM ratio was calculated. To compare hypoxia- or squalene-induced changes in individual protein levels under different conditions, data were normalized to the respective normoxic condition for each variable. To compare basal (normoxic) protein levels under different conditions, data were normalized to a single control condition. All data were obtained in $n \geq 3$ independent experiments, and visualization and statistical testing were performed using GraphPad Prism software as specified in figure legends. Grey lines on graphs indicate statistical comparisons. Where multiple statistical comparisons were made in a single experiment, p-values were corrected using the Benjamini-Hochberg method (*Benjamini and Hochberg, 1995*) with a false discovery threshold of 5%. Thresholds for statistical significance were defined as: *, $p \leq 0.05$; **, $p \leq 0.01$. Values of $0.05 < p < 0.075$ are indicated in text on graphs, and n.s. indicates $p \geq 0.075$. Figures were assembled using Adobe Illustrator software (Adobe Inc).

## Acknowledgements

We thank Dr Martin Bucknall for technical assistance with gas chromatography-mass spectrometry and the members of the Brown laboratory for critically reviewing this manuscript.

## Additional information

### Funding

| Funder | Grant reference number | Author |
| --- | --- | --- |
| Australian Government | Research Training Program Scholarship | Hudson W Coates |
| University of New South Wales | Scientia PhD Scholarship | Isabelle M Capell-Hattam |
| RANZCOG Research Foundation | Mary Elizabeth Courier Research Scholarship | Rhonda Farrell |
| Cancer Institute NSW | Career Development Fellowship | Frances L Byrne |
| Australian Research Council | Grant DP170101178 | Andrew J Brown |
| NSW Health | Investigator Development Grant | Andrew J Brown |
| Cancer Institute NSW | 2021/CDF1120 | Frances L Byrne |

The funders had no role in study design, data collection and interpretation, or the decision to submit the work for publication.

### Author contributions

Hudson W Coates, Conceptualization, Investigation, Visualization, Methodology, Writing - original draft, Writing - review and editing; Isabelle M Capell-Hattam, Investigation, Methodology; Ellen M Olzomer, Methodology; Ximing Du, Resources, Methodology; Rhonda Farrell, Hongyuan Yang, Resources, Funding acquisition; Frances L Byrne, Resources, Funding acquisition, Writing - review and editing; Andrew J Brown, Conceptualization, Resources, Supervision, Funding acquisition, Methodology, Writing - review and editing

## Author ORCIDs

Hudson W Coates http://orcid.org/0000-0002-6506-5249
Ximing Du http://orcid.org/0000-0002-6648-017X
Hongyuan Yang http://orcid.org/0000-0002-8482-6031
Andrew J Brown http://orcid.org/0000-0002-4475-0116

## Decision letter and Author response

Decision letter https://doi.org/10.7554/eLife.82843.sa1
Author response https://doi.org/10.7554/eLife.82843.sa2

## Additional files

### Supplementary files
• MDAR checklist

### Data availability

All data generated or analyzed during this study are included in the manuscript. Uncropped immuno-blots and thin-layer chromatography images are accessible as source data.

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

## Appendix 1

**Appendix 1—table 1.** Plasmids used for transfection.

| Plasmid | Description |
| --- | --- |
| pCMV-(HA)₃-SM-V5 | pcDNA3.1/V5-His TOPO vector containing the coding sequence of human squalene monooxygenase (SM; NM_003129.4) fused with three N-terminal HA epitope tags and C-terminal V5 and 6×His epitope tags, under the transcriptional control of a constitutive cytomegalovirus (CMV) promoter. Generated previously by the Brown laboratory. |
| pCMV-(HA)₃-SM-V5 K82 / 90 / 100R | pCMV-(HA)₃-SM-V5 containing SM K82R, K90R and K100R substitutions, which blunt SM truncation (**Coates et al., 2021**). Generated previously (**Coates et al., 2021**). |
| pCMV-(HA)₃-SM-V5 K290R | pCMV-(HA)₃-SM-V5 containing a K290R substitution, which disrupts a known ubiquitination site (**Hornbeck et al., 2015**). Generated in this study. |
| pCMV-(HA)₃-SM-V5 SM-N100 C/S/T>A | pCMV-(HA)₃-SM-V5 containing SM T3A, T9A, T11A, S43A, C46A, S59A, S61A, S67A, S71A, S83A and S87A substitutions, which blunt cholesterol-induced degradation of SM (**Chua et al., 2019**). Generated previously (**Coates et al., 2021**). |
| pCMV-(HA)₃-SM-V5 Y195F | pCMV-(HA)₃-SM-V5 containing an SM Y195F substitution, which causes loss of catalytic activity (**Padyana et al., 2019**). Generated previously (**Coates et al., 2021**). |
| pCMV-SM(ΔN65)-V5 | pCMV-(HA)₃-SM-V5 containing a deletion of the N-terminal (HA)₃ tag and first 65 residues of SM, which are lost upon truncation (**Coates et al., 2021**). Generated previously (**Coates et al., 2021**). |
| pCMV-SM(ΔN65)-V5 Y195F | pCMV-SM(ΔN65)-V5 containing an SM Y195F substitution, which causes loss of catalytic activity (**Padyana et al., 2019**). Generated in this study. |
| pCMV-(HA)₃-SM-V5 F20A | pCMV-(HA)₃-SM-V5 containing a F20A substitution within the N-terminal re-entrant loop (**Howe et al., 2015**). Generated in this study. |
| pCMV-(HA)₃-SM-V5 L30A | pCMV-(HA)₃-SM-V5 containing a L30A substitution within the N-terminal re-entrant loop (**Howe et al., 2015**). Generated in this study. |
| pCMV-(HA)₃-SM-V5 F35A | pCMV-(HA)₃-SM-V5 containing a F35A substitution within the N-terminal re-entrant loop (**Howe et al., 2015**). Generated in this study. |
| pCMV-(HA)₃-SM-V5 L42A | pCMV-(HA)₃-SM-V5 containing a L42A substitution within the N-terminal re-entrant loop (**Howe et al., 2015**). Generated in this study. |
| pCMV-(HA)₃-SM-V5 F20A / L30A / F35A / L42A | pCMV-(HA)₃-SM-V5 containing F20A, L30A, F35A and L42A substitutions within the N-terminal re-entrant loop (**Howe et al., 2015**). Generated in this study. |
| PX458 *SQLE*-1 (promoter) | PX458 containing a guide RNA sequence targeting the *SQLE* proximal promoter (hg38 chr8:124998048–124998067). Generated in this study. |
| PX458 *SQLE*-2 (exon 1) | PX458 containing a guide RNA sequence targeting *SQLE* exon 1 (hg38 chr8:124999588–124999607). Generated in this study. |

**Appendix 1—table 2.** Primers used for qRT-PCR and DNA cloning.

Non-annealing nucleotides for restriction enzyme cloning and DNA substitution are indicated in lowercase.

| qRT-PCR primer pair | | Primer sequence (5′–3′) | Source |
|---|---|---|---|
| *VEGF* | Forward<br>Reverse | CCTGGTGGACATCTTCCAGG<br>CTGTAGGAAGCTCATCTCTCC | *Coates et al., 2019* |
| *CA9* | Forward<br>Reverse | GTGCCTATGAGCAGTTGCTGTC<br>AAGTAGCGGCTGAAGTCAGAGG | Origene |
| *RPL11* | Forward<br>Reverse | AGAGTGGAGACAGACTGACGCG<br>CGGATGCCAAAGGATCTGACAG | Origene |
| *GAPDH* | Forward<br>Reverse | GTCTCCTCTGACTTCAACAGCG<br>ACCACCCTGTTGCTGTAGCCAA | Origene |
| *ACTB* | Forward<br>Reverse | CACCATTGGCAATGAGCGGTTC<br>AGGTCTTTGCGGATGTCCACGT | Origene |
| *HIF1A* | Forward<br>Reverse | CAGTAACCAACCTCAGTGTGG<br>CAGATGATCAGAGTCCAAAGC | Dr Laura Sharpe (Brown laboratory) |
| *EPAS1* (*HIF2A*) | Forward<br>Reverse | CTGTGTCTGAGAAGAGTAACTTCC<br>TTGCCATAGGCTGAGGACTCCT | Origene |
| *SQLE* | Forward<br>Reverse | GCTTCCTTCCTCCTTCATCAGTG<br>GCAACAGTCATTCCTCCACCA | *Coates et al., 2021* |
| *HMGCR* | Forward<br>Reverse | TTGGTGATGGGAGCTTGCTGTG<br>AGTCACAAGCACGTGGAAGACG | *Wong et al., 2007* |

| DNA cloning primer pair | | Primer sequence (5′–3′) | Method |
|---|---|---|---|
| *SQLE* gRNA 1 (promoter) | Forward<br>Reverse | caccgAATGGAAACGTTCCGACCCG<br>aaacCGGGTCGGAACGTTTCCATTc | *BbsI* cloning into PX598 vector (as in *Ran et al., 2013*) |
| *SQLE* gRNA 2 (exon 1) | Forward<br>Reverse | aaacAGTTCGCCCTCTTCTCGGATc<br>caccgATCCGAGAAGAGGGCGAACT | |
| SM K290R | Forward<br>Reverse | GATGGGCTTTTCTCCAgGTTCAGGAAAAGCCTG<br>GCGATGCAATTTCCTCATTT | Overlap extension cloning (as in *Stevenson et al., 2013*) |
| SM Y195F | Forward<br>Reverse | CCAGGTTGTAAATGGTTtCATGATTCATGATCAGG<br>GCGATGCAATTTCCTCATTT | |
| SM F20A | Forward<br>Reverse | GTTCGGGGACgcCATCACTTTGG<br>GCGATGCAATTTCCTCATTT | |
| SM L30A | Forward<br>Reverse | GGAGGTCCTGgcGTGCGTGCTGGTG<br>GCGATGCAATTTCCTCATTT | |
| SM F35A | Forward<br>Reverse | GTGCTGGTGgcCCTCTCGCTG<br>GCGATGCAATTTCCTCATTT | |
| SM L42A | Forward<br>Reverse | CTGGGCCTGGTGgcCTCCTACCGCTG<br>GCGATGCAATTTCCTCATTT | |
| SM F20A / L30A / F35A / L42A | Forward<br>Reverse | Combination of above mutagenic primers<br>GCGATGCAATTTCCTCATTT | |

