## [Editor Report]

Cholesterol biosynthesis is a highly oxygen-intensive process as the synthesis of one molecule of cholesterol consumes 11 molecules of oxygen. This valuable paper provides a new link between oxygen sensing and cholesterol synthesis by showing that under conditions of hypoxia (oxygen deprivation), a key cholesterol synthesis enzyme called squalene monooxygenase (SM) is partially degraded to a truncated form that is constitutively active. The supporting evidence is solid and suggests that unregulated activation of SM under oxygen-deficient conditions could reduce the toxicity of squalene and other sterol intermediates.

---

## [Decision Letter]

**Decision letter after peer review:**

Thank you for submitting your article "Hypoxia truncates and constitutively activates the key cholesterol synthesis enzyme squalene monooxygenase" for consideration by *eLife*. Your article has been reviewed by 3 peer reviewers, and the evaluation has been overseen by a Reviewing Editor and David Ron as the Senior Editor. The reviewers have opted to remain anonymous.

Essential revisions:

Overall, the reviewers were positive about this work and felt that the induction of truncated squalene monooxygenase (truncSM) by hypoxia (relieving it from cholesterol-mediated degradation) was a convincing and exciting result. This result is the core of the manuscript and should be shored up as much as possible. To improve the clarity and presentation, the reviewers suggest some key revisions, which are listed below. Please address these in full and also address the other points raised in the detailed reviews which are appended below.

1) All reviewers were concerned/confused by some of the immunoblot quantification. The interpretations would be simplified if the authors were to show results of quantification for each band in the immunoblots (higher quality immunoblots should be presented if possible). In many cases the graphs of the quantifications seem to be off, it is not clear how the graphs are derived from the blots, which should be representative – see further comments by Reviewers 2 and 3 below on this point.

2) The MARCH6 connection is preliminary/confusing and the inhibitor studies are not clear. A new band appears upon inhibition of VCP/p97 and MG-132 treatment causes a decrease in the protein's expression. This needs to be addressed. The interpretation of Figure 2E is somewhat overstated. Please also see other points about MARCH6 experiments raised below. If it cannot be clarified, the reviewers felt that the MARCH6 angle could be significantly toned down (with some panels removed) without detraction from the main point about hypoxia inducing truncSM.

3) Finally, results shown in Figure 5 showing that truncation of SM correlates with hypoxia in endometrial cancer tissues is a little preliminary. Multiple bands are detected in SM immunoblots, which interferes with interpretation. This experiment could be removed and speculated upon in the discussion.

*Reviewer #1 (Recommendations for the authors):*

I do have a couple of expository suggestions.

1) You have GOT to include a picture or schematic of the sterol pathway with the relevant enzymes. I have been studying aspects of this pathway for over 30 years (I am old) and I still need one. The people who are new to these ideas will be even more appreciative of this graphic concession.

2) Can you make some kind of summary cartoon of the various regulation processes happening with this key enzyme? I am not sure what it would look like, but it would be a nice summary of the complex and cool story.

*Reviewer #2 (Recommendations for the authors):*

Is it a clear accumulation of truncSM that results in increased flux through the pathway and sterol metabolic changes? If yes, does this provide a significant advantage to cells under hypoxia? The accumulation of squalene in hypoxia argues against significant truncSM activity. Even if truncSM can be active, oxygen unavailability possibly limits its activity. This possibility is in fact acknowledged by the authors in Line 313 ("consistent with reduced SM activity under low-oxygen conditions").

Another important open question is the role of squalene in promoting truncSM. Under normoxia, MARCHF6 induces the degradation of full-length SM. How is the generation of truncSM favoured by squalene? Does it act directly on MARCHF6? Any additional information to address these issues would significantly strengthen this study. The analysis and some of the data on the relative abundance of SM and truncSM could also be improved.

1. There is huge variability between the control condition (cells grown at 21% O2). For example, in Figure 1D, at .5% O2 the ratio between SM/truncSM is big (probably >5). In contrast, at 3% or 10% O2 the SM/truncSM ratio is probably ~1. It would be important to understand the causes of variability that are observed throughout the manuscript.

2. Another important aspect is the quantification and presentation of the data. The levels of SM and truncSM at 21%O2 are used as references and the two versions of the protein are quantified separately. This is misleading because often cells have different levels of total protein (SM+truncSM) (see for example figure 2). Throughout the manuscript, the levels of truncated protein should be presented in a ratio to full length to provide a better indication of their relative abundance. Another example of strange quantification is in Figures 2C (and 2E). The gel on the left shows very high levels of SM-N100-GFP-V5 in cells treated with MG132. However, in the quantification on the right, those conditions appear to have less SM-N100-GFP-V5 than untreated cells under 21% O2.

3- Figure 2E shows a clear increase in truncSM in MARCHF6 depleted cells. The effect is not insignificant if the truncSM/SM ratio is analysed. In contrast, the effect of MARCHF6 depletion is clear on full-length accumulation. Is this result compatible with the conclusion that "truncation occurs post-ubiquitination by MARCHF6"?

*Reviewer #3 (Recommendations for the authors):*

The availability of data, code, reagents and other issues is adequate.

---

## [Author Response]

Essential revisions:Overall, the reviewers were positive about this work and felt that the induction of truncated squalene monooxygenase (truncSM) by hypoxia (relieving it from cholesterol-mediated degradation) was a convincing and exciting result. This result is the core of the manuscript and should be shored up as much as possible. To improve the clarity and presentation, the reviewers suggest some key revisions, which are listed below. Please address these in full and also address the other points raised in the detailed reviews which are appended below.1) All reviewers were concerned/confused by some of the immunoblot quantification. The interpretations would be simplified if the authors were to show results of quantification for each band in the immunoblots (higher quality immunoblots should be presented if possible). In many cases the graphs of the quantifications seem to be off, it is not clear how the graphs are derived from the blots, which should be representative – see further comments by Reviewers 2 and 3 below on this point.

Clarifying our quantification method is clearly essential. Below we address the specific recommendations of Reviewers #2 and #3, and the revised manuscript incorporates these where possible. However, in some cases, we believe alternative quantification methods would distract from our aim to identify factors required for hypoxia-induced SM degradation and trunSM accumulation. We have nevertheless made major updates to the text, figure legends, and axis labels to improve clarity, as outlined below.

Presenting data for all immunoblot lanes: We agree this quantification method, suggested by Reviewer #3, is informative in situations where a condition dramatically affects protein levels at the normoxic baseline. See Author response image 1 for an example based on the original Figure 2C, where normalizing to a single control condition (the normoxic vehicle treatment, set to 1) clearly demonstrates the MG132-induced accumulation of SM and SM‑N100‑GFP‑V5, as seen in the accompanying immunoblot.

**Author response image 1. sa2fig1:** 

However, a limitation of this normalization method is that the magnitude of hypoxia-induced changes in protein levels cannot be directly compared across multiple treatments or conditions. This was the principle we used to investigate the mechanism of SM truncation, and the rationale behind our original quantification method where protein levels were normalized to the normoxic control for each individual treatment (set to 1). We acknowledge our original visualization of this data, with a single bar representing all normoxic conditions in an experiment (see Author response image 2), was confusing and potentially misleading. The main figures of the revised manuscript still contain quantification of hypoxia-induced protein changes, but the graphs are updated to show all normoxic conditions used for normalization, and thus provide a numerical value for each immunoblot lane. See Author response image 2 for an example of the revised Figure 2C (now Figure 3A) compared with the quantification in the original submission. Where treatments cause normoxic protein accumulation with the potential to confuse interpretation, the supplemental data is updated to include quantification relative to a single control condition, as in the example on the previous page. We have elected to include this additional quantification for the revised Figure 2C (as Figure 2—figure supplement 1C), Figure 2D (as Figure 2—figure supplement 1E) Figure 3A (as Figure 3—figure supplement 1B), and Figure 4—figure supplement 2E, which were flagged by reviewers as points of particular concern.

We have also updated axis labels and figure legends to improve clarity about the normalization process. For instance, the above graphs replace the original “relative protein levels” label with a more informative “hypoxia-induced reduction” label. The Materials and methods section is also updated to better describe the quantification methods used in the manuscript.

Presenting the trunSM:SM ratio: This quantification method, suggested by Reviewer #2, is useful for visualizing the overall shift from full-length to truncated SM during hypoxia, as was the focus of Figure 1. Expressing the data as a ratio increases variability, but nevertheless supports our original conclusions from these experiments. See Author response image 3 for an example of trunSM:SM quantification from Figure 1D, which Reviewer #2 flagged as a concern due to differences in the normoxic ratio. We also note that cell confluency influences the trunSM:SM ratio—for more information, please refer to our response to Reviewer #2, point #1. The revised manuscript incorporates trunSM:SM quantification for all Figure 1 experiments as Figure 1—figure supplement 2B–D, F and Figure 1—figure supplement 3B.

**Author response image 3. sa2fig3:** 

When investigating the mechanism of truncation in Figure 2 onwards, we believe presenting the trunSM:SM ratio risks obscuring factors that specifically affect only one of the two steps in the truncation process: (1) proteasomal targeting and (2) partial degradation. This is best illustrated by Figure 4A, which uses different serum types. Here, individual quantification of the two SM variants clearly shows that statin treatment blocks the hypoxia-induced generation of trunSM without an effect on the disappearance (i.e., proteasomal targeting) of full-length SM (Author response image 4, left). This observation was critical, as it led directly to our discovery that squalene promotes the partial, rather than complete, degradation of SM. However, there remains a significant increase in the trunSM:SM ratio in the presence of statin due to the continued disappearance of full-length SM (Author response image 4, red box). Expressing the data in this way makes the differential effects on each SM variant much less apparent, and we believe it may lead to confusion about our conclusions. In the revised manuscript, the Materials and methods section is updated to justify our decisions regarding this quantification method.

**Author response image 4. sa2fig4:** 

2) The MARCH6 connection is preliminary/confusing and the inhibitor studies are not clear. A new band appears upon inhibition of VCP/p97 and MG-132 treatment causes a decrease in the protein's expression. This needs to be addressed. The interpretation of Figure 2E is somewhat overstated. Please also see other points about MARCH6 experiments raised below. If it cannot be clarified, the reviewers felt that the MARCH6 angle could be significantly toned down (with some panels removed) without detraction from the main point about hypoxia inducing truncSM.

We accept that some of our MARCHF6 experiments are preliminary, and the two MARCHF6‑V5 bands confound their interpretation. The revised manuscript contains only the *MARCHF6* knockdown experiment in Figure 2E (showing MARCHF6 is required for hypoxia-induced SM degradation) and the observation of hypoxic MARCHF6 stabilization in Figure 3A. These results are merged into an updated Figure 3. Experiments to determine the mechanism of MARCHF6 stabilization (the original Figure 3B–C, Supplemental Figure S3, Supplemental Figure S5F) are removed and will be the focus of a future study. The working model in Figure 6 and results and discussion sections are revised accordingly.

3) Finally, results shown in Figure 5 showing that truncation of SM correlates with hypoxia in endometrial cancer tissues is a little preliminary. Multiple bands are detected in SM immunoblots, which interferes with interpretation. This experiment could be removed and speculated upon in the discussion.

While the correlation between HIF1α levels and SM truncation in endometrial cancer tissues supports our cell culture work, we acknowledge it could be seen as somewhat preliminary. The original Figure 5 is removed from the revised manuscript and the Discussion section is revised to highlight the role of trunSM in SM-related oncogenesis as a future direction for study.

Reviewer #1 (Recommendations for the authors):I do have a couple of expository suggestions.1) You have GOT to include a picture or schematic of the sterol pathway with the relevant enzymes. I have been studying aspects of this pathway for over 30 years (I am old) and I still need one. The people who are new to these ideas will be even more appreciative of this graphic concession.

As suggested, the revised manuscript includes a schematic of the cholesterol synthesis pathway highlighting key enzymes mentioned in the study. This is incorporated as Figure 1—figure supplement 1.

2) Can you make some kind of summary cartoon of the various regulation processes happening with this key enzyme? I am not sure what it would look like, but it would be a nice summary of the complex and cool story.

As suggested, the revised manuscript includes a graphical summary of metabolic regulation of SM. This is incorporated as Figure 6—figure supplement 1 and the Discussion section is updated accordingly.

Reviewer #2 (Recommendations for the authors):Is it a clear accumulation of truncSM that results in increased flux through the pathway and sterol metabolic changes? If yes, does this provide a significant advantage to cells under hypoxia? The accumulation of squalene in hypoxia argues against significant truncSM activity. Even if truncSM can be active, oxygen unavailability possibly limits its activity. This possibility is in fact acknowledged by the authors in Line 313 ("consistent with reduced SM activity under low-oxygen conditions").

We agree it is important to determine if trunSM activity is preserved during hypoxia. In the revised manuscript, we use [^14^C]‑acetate labelling and thin-layer chromatography to assay flux through cholesterol synthesis in hypoxic Huh7 cells. This liver-derived cell line shows robust hypoxia-induced truncation of SM (Figure 1—figure supplement 2F). While cholesterol synthesis is reduced during hypoxia, we observed acute accumulation of lanosterol (consistent with other reports, e.g., Nguyen *et al.* 2007 *JBC*) yet minimal upstream accumulation of squalene even after a prolonged incubation. This confirms that despite trunSM being oxygen-dependent, its catalytic activity is preserved during hypoxia. We posit that a major function of SM truncation in this context is to ensure efficient clearance of squalene to form lanosterol, which induces degradation of the early cholesterol synthesis enzyme HMGCR and thus suppresses oxygen-intensive pathway flux.

Although we detected squalene accumulation at early hypoxic timepoints in earlier gas chromatography-mass spectrometry experiments (Figure 4B), we attribute this to the much greater sensitivity of mass spectrometry compared with thin-layer chromatography. Indeed, squalene levels are at the lower level of detection in normoxic cells (Figure 4—figure supplement 1) and the strong correlation between trunSM and squalene levels, as determined by mass spectrometry, points towards truncation responding to extremely small changes in squalene levels. The revised manuscript incorporates the new thin-layer chromatography experiment as Figure 5, and the working model in Figure 6 and results, discussion and Materials and methods sections are updated accordingly. Please note the addition of a new author, Isabelle M. Capell-Hattam, who contributed to these experiments.

Another important open question is the role of squalene in promoting truncSM. Under normoxia, MARCHF6 induces the degradation of full-length SM. How is the generation of truncSM favoured by squalene? Does it act directly on MARCHF6? Any additional information to address these issues would significantly strengthen this study.

As suggested, the revised manuscript tests another possible mechanism by which squalene induces SM truncation. Previous work showed squalene directly binds the SM‑N100 regulatory domain (Yoshioka *et al.* 2020 *PNAS*), with the hydrophobic re-entrant loop (residues ~15–40) as the most logical binding site. Aromatic residues and leucine residues line the SM active site and are required for catalysis (Padyana *et al.* 2019 *Nat Commun*, Abe *et al.* 2007 *BBRC*), suggesting they directly interact with squalene. Similar residues in the SM‑N100 re-entrant loop may likewise be involved in squalene binding, and possibly the partial degradation of SM. We therefore mutated phenylalanine (Phe‑20, Phe‑35) and leucine (Leu‑30, Leu‑42) residues in the re-entrant loop to test if they are required for SM truncation. However, these mutations, either alone or in combination, did not prevent the hypoxia-induced accumulation of truncated SM. A better understanding of the precise squalene-binding site in SM‑N100 is needed to further investigate its possible role in SM truncation, but this is beyond the scope of the current study. The revised manuscript incorporates this new SM mutant experiment as Figure 4—figure supplement 4, and the results and discussion sections are updated accordingly.

The analysis and some of the data on the relative abundance of SM and truncSM could also be improved.

Please refer below for our responses to specific concerns, and to our response to Essential Revisions comment #2 for overall changes to data normalization and visualization.

1. There is huge variability between the control condition (cells grown at 21% O2). For example, in Figure 1D, at .5% O2 the ratio between SM/truncSM is big (probably >5). In contrast, at 3% or 10% O2 the SM/truncSM ratio is probably ~1. It would be important to understand the causes of variability that are observed throughout the manuscript.

Due to equipment limitations affecting the number of oxygen concentrations testable at one time, experiments in Figure 1D were staggered across multiple days after cell seeding. This meant that some cells were incubated for longer than others prior to their hypoxic exposure. While steps were taken to obtain a similar cell density at the start of each hypoxic incubation, unavoidable differences in confluency may explain the variability in basal trunSM:SM ratios. We have formally tested the effect of HEK293T seeding density on SM truncation and found a higher confluency indeed promotes a greater trunSM:SM ratio. This is accompanied by a small increase in HIF1α levels that likely reflects increased oxygen consumption of confluent cell monolayers, as reported by other studies (Sheta *et al.* 2001 *Oncogene*, Dayan *et al.* 2009 *JCP*). Therefore, this observation lends additional support to our model of hypoxia-induced truncation. In other manuscript figures, differences in the basal trunSM:SM ratio likely reflect variation amongst HEK sublines (e.g., HEK293T in Figure 1A, HEK SM‑N100‑GFP‑V5 in Figure 2C, and HEK MARCHF6‑V5 in Figure 3B). The revised manuscript incorporates the new cell confluency experiment as Figure 1—figure supplement 2D, and the Results section is updated accordingly. To improve readability, the original Supplemental Figure S1D–E is moved to a new Figure 1—figure supplement 3.

2. Another important aspect is the quantification and presentation of the data. The levels of SM and truncSM at 21%O2 are used as references and the two versions of the protein are quantified separately. This is misleading because often cells have different levels of total protein (SM+truncSM) (see for example figure 2). Throughout the manuscript, the levels of truncated protein should be presented in a ratio to full length to provide a better indication of their relative abundance. Another example of strange quantification is in Figures 2C (and 2E). The gel on the left shows very high levels of SM-N100-GFP-V5 in cells treated with MG132. However, in the quantification on the right, those conditions appear to have less SM-N100-GFP-V5 than untreated cells under 21% O2.

The revised manuscript includes quantification of the trunSM:SM ratio for experiments in Figure 1, and quantification of protein levels for all immunoblot lanes, including in Figure 2C. It also contains updates to the text, figure legends, and axis labels to improve clarity about data normalization. For more information, please refer to our response to Essential Revisions comment #1.

3- Figure 2E shows a clear increase in truncSM in MARCHF6 depleted cells. The effect is not insignificant if the truncSM/SM ratio is analysed. In contrast, the effect of MARCHF6 depletion is clear on full-length accumulation. Is this result compatible with the conclusion that "truncation occurs post-ubiquitination by MARCHF6"?

The reviewer is correct that hypoxia-induced trunSM accumulation still occurs in *MARCHF6*-depleted cells. This is also reflected in the trunSM:SM ratio, which remains significantly increased (Author response image 5). Our interpretation is that MARCHF6 is not the sole E3 ubiquitin ligase promoting trunSM formation, and although it is responsible for hypoxia-induced degradation of SM, other proteasomal degradation pathways can cause partial degradation of SM when hypoxia-induced squalene accumulation occurs. To better convey this idea, we have modified our conclusion as follows: “The basal levels and hypoxic accumulation of trunSM were also reduced, consistent with MARCHF6 contributing to the proteasomal targeting, and therefore partial degradation, of SM. Hypoxia-induced accumulation of trunSM was not completely abolished, however, indicating SM can be truncated even when targeted to the proteasome by hypoxia-independent mechanisms.”

**Author response image 5. sa2fig5:** 

Reviewer #3 (Recommendations for the authors):The availability of data, code, reagents and other issues is adequate.

The revised manuscript includes availability information for all new experiments.